

# Novel Arctic sea ice data assimilation combining ensemble Kalman filter with a Lagrangian sea ice model

Sukun Cheng[1], Yumeng Chen[2], Ali Aydoğdu[3], Laurent Bertino[1], Alberto Carrassi[2,4], Pierre Rampal[5], and Christopher K. R. T. Jones[6]

[1]Nansen Environmental and Remote Sensing Center, 5007 Bergen, Norway
[2]Department of Meteorology and National Centre for Earth Observation, University of Reading, Reading RG6 6AH, UK
[3]Ocean Modelling and Data Assimilation Division, Fondazione Centro Euro-Mediterraneo sui Cambiamenti Climatici (CMCC), 40127, Bologna, Italy
[4]Department of Physics and Astronomy "Augusto Righi", University of Bologna. Bologna, Italy
[5]Institut de Géophysique de l'Environnement, Université Grenoble Alpes/CNRS/IRD/G-INP, CS 40700, 38 058 Grenoble CEDEX 9, France
[6]Department of Mathematics, University of North Carolina, Chapel Hill, USA

**Correspondence:** Sukun Cheng (sukun.cheng@nersc.no)

**Abstract.** Advanced data assimilation (DA) methods, widely used in geophysical and climate studies to merge observations with numerical models, can improve the state estimates and consequent forecasts. We interface the deterministic Ensemble Kalman filter (DEnKF) to the Lagrangian sea ice model, neXtSIM. The ensemble is generated by perturbing the atmospheric and oceanic forcing throughout the simulations and randomly initialized ice cohesion. Our ensemble-DA system assimilates sea ice concentration (SIC) from the Ocean and Sea Ice Satellite Application Facility (OSI-SAF) and sea ice thickness (SIT) from the merged CryoSat-2 and SMOS datasets (CS2SMOS). Because neXtSIM is computationally solved on a time-dependent evolving mesh, it is a challenging application for ensemble DA. As a solution, we perform the DEnKF analysis on a fixed reference mesh, where model variables are interpolated before the DA and then back to each member's mesh after the DA. We evaluate the impact of assimilating different types of sea-ice observations on the model's forecast skills of the Arctic sea ice by comparing against satellite observations and a free-run ensemble in an Arctic winter period, 2019-2020. Significant improvements in modeled SIT indicate the importance of assimilating weekly CS2SMOS SIT, while the improvement of SIC and ice extent are moderate but benefit from daily ingestion of the OSI-SAF SIC. In contrast, the bivariate improvements between SIC and SIT are unobvious. Our ensemble-DA system based on the stand-alone sea ice model is computationally efficient and demonstrates comparable skills to operational forecasting models that use DA.

## 1 Introduction

Sea ice is a critical component of the Earth system. The evident loss of Arctic sea ice both as sea ice extent (SIE) and volume (SIV) over recent decades has been abundantly discussed(e.g., Meier, 2017). Predicting the Arctic sea ice conditions from near term to decadal timescales becomes increasingly important due to their importance both for the Earth's climate and local human activities such as shipping that require accurate sea ice forecast (Bertino and Holland, 2017; Wagner et al., 2020).





Like other climate system components, sea ice forecasting relies on numerical models and observational data. Sea ice models are sensitive to both initial and boundary conditions, which can be constrained by observations using data assimilation (DA, e.g., Carrassi et al., 2018). Satellite observations are critical in polar regions that offer a homogeneous and dense spatial and temporal coverage, in contrast, the in-situ measurements are relatively sparse and generally collected in the summer. Satellite-based observations of sea ice concentration (SIC) have been available since the 1970s and have been widely used to

calibrate coupled ocean-ice models (Lisæter et al., 2003; Stark et al., 2007; Posey et al., 2015). Those studies have shown that assimilating SIC alone with a multivariate DA scheme improves the short-term forecast of SIC, sea ice thickness (SIT), and sea surface temperature (SST) (Lisæter et al., 2003; Massonnet et al., 2015). Furthermore, assimilation of SIC in a coupled climate model was also beneficial to seasonal scales (Kimmritz et al., 2019). Contrary to SIC data, the satellite record of SIT products is shorter. Although these products are still affected by significant errors, satellite-retrieved SIT measurements have

been nevertheless assimilated in recent years with some success (Xie et al., 2018; Allard et al., 2018; Fritzner et al., 2019). For example, Xie et al. (2018) assimilated the SIT observations of CS2SMOS (Ricker et al., 2017), which combines observations from Cryosat-2 altimeter (Laxon et al., 2013) and the Soil Moisture and Ocean Salinity (SMOS) radiometer (Kaleschke et al., 2016), into TOPAZ4 (version 4 of a coupled ocean-sea ice data assimilation system for the North Atlantic and Arctic, (Sakov et al., 2012)). They recommended the assimilation of SIT to improve SIT and sea ice drift (SID) in the reanalysis. Fritzner

et al. (2019) found that assimilating SIT improved the prediction of SIT, SIC, and snow depth in the coupled ocean-ice model ROMS-CICE, which is composed of the Regional Ocean Modeling System (ROMS, Shchepetkin and McWilliams (2005)) and the Los Alamos sea-ice model (CICE, Hunke et al. (2010)).

    This study presents a novel application of the ensemble Kalman filter (EnKF, Evensen, 2003) used to assimilate satellite-based SIC and SIT data in the Lagrangian neXt generation Sea Ice Model (neXtSIM, Rampal et al., 2016b, 2019). Our work

builds upon and extends the preliminary DA study with neXtSIM from Williams et al. (2021). They introduced the deterministic forecasting platform neXtSIM-F whereby the OSI-SAF SIC observations (both brightness temperatures measured by the Special Sensor Microwave Imager Sounder - SSMIS and Advanced Microwave Scanning Radiometer 2 - AMSR2) were assimilated by a simple DA method - "direct insertion". Improvements in forecasting SIE were substantial and have been used for short-term operational forecasting as part of the Copernicus Marine Services. Following this line, we study here the impact on

sea ice forecast skill determined by a state-of-the-art ensemble-DA method, the deterministic EnKF (DEnKF; Sakov and Oke, 2008). The advantages over "direct insertion" are multiple, including the account of both model and observational errors when computing the analysis, its multivariate character such that observations influences are not limited to their spatial locations and measured variable, as well as its provision of an ensemble of model trajectories prone to probabilistic predictions. The multivariate aspect will not appear in the present study because the observations assimilated have complete spatial coverage

and observe the two most important model variables (SIC and SIT). Nevertheless, this issue will be important when performing ensemble-DA with neXtSIM in a coupled model.

    In this study, the ensemble for the DEnKF is constructed by simultaneously perturbing model external forcing and one of its internal parameters, following the sensitivity studies and probabilistic predictions from Rabatel et al. (2018) and Cheng et al. (2020). In particular, Rabatel et al. (2018) demonstrated that the external atmospheric forcing is the primary driver of





prediction uncertainty in neXtSIM and thus controls the diversity of the ensemble members: predicting sea ice is more of a boundary condition than an initial value problem. In previous studies, neXtSIM used the Elasto-Brittle and Maxwell-Elasto-Brittle (MEB) rheology for the sea-ice internal stress, whose main control parameter is the ice cohesion, determining sea ice damage. As shown in Cheng et al. (2020), it is convenient to perturb the (internal) ice cohesion together with the (external) wind forcing to enhance the ensemble dispersion. Although we use the latest version of neXtSIM that employs a slightly different sea ice rheology (see 2.1), we can still adopt their strategy to construct ensemble.

The neXtSIM model is solved on a Lagrangian mesh which nodes move following the ice velocity fields and are periodically removed/added at "remeshing" steps to keep the mesh geometry within prescribed tolerances. Consequently, the mesh node locations and their total numbers change with time, which differ between ensemble members. These numerical features, increasingly present in the wider computational physics area (e.g. Alam and Lin, 2008), render the application of EnKF-like methods cumbersome: particular adaptations of the algorithm are needed. Although Lagrangian vertical coordinates in an ocean model can be handled correctly (Wang et al., 2016), the horizontal two-dimensional problem is more complex because there is no unique ordering of the mesh. Here we use the approach introduced in 1D by Aydoğdu et al. (2019), which uses a fixed-in-time (possibly homogeneous) reference mesh to perform the analysis updates. In practice, at each analysis step, the individual ensemble members are projected onto the reference mesh where the algebra of the analysis update is performed and then projected back on the individual meshes (unaltered by the analysis). The method was further developed by Sampson et al. (2021) so that the individual mesh of each ensemble member is also informed at the analysis steps: this led to better predictions, particularly in the proximity of steep gradients in the physical quantities. In both cases, the fixed reference mesh (and its upper/lower resolution bounds) is chosen based on the aforementioned geometric tolerances of the model mesh, then kept fixed and used throughout the entire duration of the experiments. For simplicity, in the present 2D neXtSIM context, we adopt the method originally proposed by Aydoğdu et al. (2019).

The paper is organized as follows. Section 2.1 describes the neXtSIM model and its setup. It follows with a description of the observation products used in this study in Section 3. Section 4 presents the ensemble-DA framework for neXtSIM with the DEnKF for sea ice forecasting and a brief introduction of DEnKF. Section 5 describes the experiment setup. In Section 6 the resulting sea-ice quantities are evaluated among different DA strategies. Discussion and conclusions are given in Section 7.

## 2 Numerical model and configuration

### 2.1 neXtSIM

The neXtSIM model is a full dynamic/thermodynamic sea ice model (Rampal et al., 2016b, 2019). The latest version of neXtSIM uses a Brittle Bingham-Maxwell (BBM) rheology (Olason et al., 2022) that corresponds to a combination of the Bingham-Maxwell constitutive model (Bingham, 1922) and the Maxwell-Elasto-Brittle rheology (Dansereau et al., 2016). This rheology controls how sea ice mechanically responds to applied external forces, mainly winds and currents. The neXtSIM model has been extensively evaluated against several sets of observations of sea ice concentration, thickness, drift and deformation, and shows remarkable performances (Rampal et al., 2016a, b; Rabatel et al., 2018; Rampal et al., 2019; ?, e.g.). The



model equations are solved on an adaptive Lagrangian triangular mesh using a finite element method with remeshing to avoid extreme mesh distortion. Using a Lagrangian mesh saves computational cost from the stiff advection processes and helps pre-
serve the sharp and realistic gradients in sea ice fields. Such gradients emerge from the dynamical behavior obtained with the BBM rheology, which corresponds to the formation of leads and ridges, for instance. SIC, SIT, snow thickness, sea ice damage, sea ice velocity, and sea ice stress are included as prognostic variables (Rampal et al., 2019; **?**). Note that SIT in neXtSIM is defined as the sea ice volume divided by the grid-cell area, and it is commonly denoted as "effective sea ice thickness". The neXtSIM model also incorporates thermodynamic processes among the mechanisms responsible for ice formation and melting.
The model includes three "ice" categories: open water, young ice and old ice. The young ice category represents newly formed ice, and presuming that ice can only be transferred from the young to the old ice category. Note that the young and old ice here correspond to the thin and thick ice categories in Rampal et al. (2019), respectively. The ice categories are renamed in the latest model version for a more accurate description because the ice categories are classified by stages of sea ice development rather than ice thickness).
Sea ice processes are influenced by the atmospheric and ocean boundary conditions. It is worth noting that neXtSIM is used in a stand-alone (or uncoupled) sea-ice configuration in this study, whereby a slab ocean layer beneath the ice accounts for SST and sea surface salinity (SSS); obviously, this does not constitute a full ocean model. The SST and SSS are prognostic variables of neXtSIM updated in the exchange of heat and salinity fluxes by the thermodynamics, constrained by the ocean data via the Newtonian nudging. Thickness of the slab ocean layer is assigned as the local mixed layer depth of the ocean data.

## 2.2 Model setup

We run the neXtSIM model on a Lagrangian mesh with a resolution of about 7.5 km covering the Arctic. This resolution corresponds to a nominal triangle side length of 10 km, where the distance from one vertex of a triangle to its opposite side is 7.5 km. The time-step of the model is 900 s. The ocean bathymetric data is adopted from the ETOPO2 (2 Arc-Minute Global Relief Model) (National Geophysical Data Center, NESDIS, NOAA, U.S. Department of Commerce, 2001) for the basal stress
parameterization on sea ice (Lemieux et al., 2015).

The model is forced by the atmospheric analyses from the European Centre for Medium-Range Weather Forecasts (ECMWF) Integrated Forecasting System (IFS) operational product (0.1 degree, 9 km resolution) (Owens and Hewson, 2018), including the 10 m wind velocity, 2 m air temperature and 2 m dew point temperature, mean sea level pressure, downward long/short wave radiations, and total precipitation/snowfall. Data are projected on a polar stereographic grid every 6 hours. The forcing
data are interpolated onto the model mesh linearly in time and bi-linearly in space at run time.

The ocean forcing is from the TOPAZ4 operational analyses from the Copernicus Marine Services. The product has a horizontal resolution of 12.5 km. Specifically, we use daily analyses produced at 00:30 UTC each day. neXtSIM uses the sea surface height, near-surface ocean velocity at 30 m depth, the mixed layer depth, SST, and SSS from TOPAZ4 product, which is the average of a 100-members ensemble interpolated onto the model mesh linearly in time and bi-linearly in space at run
time.



It is worthy to note that the model is forced by analyzed atmospheric and ocean data and therefore does not provide forecasts in the operational sense. However, we will hereafter refer to "forecast skills" of the model in the DA vocabulary, and call the last model run before the DA analysis by "forecast" .

As mentioned in Section 2.1, neXtSIM adjusts the SST and SSS according to the heat fluxes across the atmosphere-ice-ocean and is nudged toward the TOPAZ4 forecast product. The SST and SSS strongly influence the sea ice properties and thus can severely affect the DA outcome. The relaxation timescales are tuned using 3-month long runs to minimize the SIC error between the model runs and the OSI-SAF SIC observations (not shown). By testing the relaxation timescales from 5 to 60 days, we found that 5 days provided the best agreement with OSI-SAF SIC observations, which is smaller than the nudging timescale (15 days) in Williams et al. (2021).

## 3 Observations

We assimilate the CS2SMOS SIT observations and the European Organisation for the Exploitation of Meteorological Satellites (EUMETSAT) OSI-SAF SIC observations. In the interest of studying the feasibility of the DEnKF with neXtSIM, and because of the lack of proper independent pan-Arctic observations for validation, we primarily use the same observations products that are assimilated. As a complement, we shall also use the OSI-SAF SID as an independent validation dataset.

### 3.1 CS2SMOS sea ice thickness for assimilation

The CS2SMOS provides a daily SIT product that merges the weekly Cryosat-2 altimeter data with the daily Soil Moisture and Ocean Salinity (SMOS) radiometer data. The Cryosat-2 SIT is retrieved from the freeboard (the ice/snow height above sea level) measured by altimetry satellites, which can only detect thick ice. In contrast, the SMOS SIT is more reliable for thin ice. It is derived from microwave brightness temperatures in the L-band. The two datasets are projected on the Equal-Area Scalable Earth 2 (EASE2) 25 km grid (Brodzik et al., 2012) by optimal interpolation. The CS2SMOS data provide mapping errors as its uncertainty estimates, including the uncertainty of ice thicker than 1 m from Cryosat-2 and the uncertainty of ice thinner than 1 m from SMOS. Thick multi-year ice usually shows high uncertainty. The existence of sea ice is marked by the OSI-SAF ice-type product in CS2SMOS. We use the reprocessing mode CS2SMOS version 203 product, which uses the OSI-SAF OSI-430-b ice concentration product to identify grid cells filled with more than 15% of ice and separate the first-year ice from multi-year ice. The dataset is available for the Arctic winter from October to April, whereas it is unavailable for the melting summer season due to wet surface interference in both Cryosat-2 and SMOS.

As a merged dataset, the CS2SMOS is subject to different interpretations because Cryosat-2 provides the absolute thickness over the ice cover, defined as the sea ice volume divided by the ice-covered area fraction in the grid cell. In contrast, SMOS provides the "effective" thickness, similar to the state variable used in neXtSIM for SIT. In some studies (e.g., Mu et al., 2018), CS2SMOS data has been treated as an effective thickness with the awareness of potential uncertainties. In this study, we interpret CS2SMOS as an absolute thickness estimate (Ricker, 2021, personal communication).





## 3.2 OSI-SAF sea ice concentration for assimilation

The EUMETSAT OSI-SAF includes daily SIC products retrieved from the SSMIS data (OSI-401-b) and AMSR2 data (OSI-408) (Tonboe et al., 2017). We use the OSI-401-b product for assimilation and validation, which has a daily averaged SIC coverage under a polar stereographic grid with a horizontal resolution of 10 km.

## 3.3 OSI-SAF sea ice drift for validation

As an independent dataset, we use the OSI-SAF OSI-405 SID product to validate the SID forecast. The product is a daily product obtained from the measurements of passive and active microwave instruments that have a horizontal resolution varying from 10 to 15 km in the Arctic. The SID is retrieved by the continuous maximum cross-correlation method (Lavergne et al., 2010) and is provided on a polar stereographic regular grid of 62.5 km resolution every 48 hours. We only validate the model ice drift against this product where uncertainties on the observation-based estimates are lower than 2.5 km/2days in the winter period. The discarded high-uncertainty observations are primarily located near the north pole due to fewer observations and near the ice edge where ice drifts so fast that it reaches the limit of which the continuous maximum cross-correlation method can handle.

## 4 Ensemble-based data assimilation system

In this section, we describe the components of the ensemble-DA system: the ensemble perturbations, the statistical models of the observational errors (i.e., its error covariances), selected model states passed to the DA and the analysis scheme (DEnKF). A schematic flow chart of the system is shown in Fig. 1.

### 4.1 Perturbations of atmospheric and ocean forcing

As a crucial component of the ensemble-DA system, the forecast ensemble is generated by neXtSIM through randomly perturbed atmospheric forcing and model parameters (ice cohesion in the present case), plus randomly perturbed oceanic forcing. The former two perturbations have been investigated in Cheng et al. (2020), which shows that although the sea ice model is nonlinear, the modeled SID is less sensitive to the initial model state than to the atmospheric forcing applied throughout the run. In addition, uncertainties from the ocean processes (the TOPAZ4 data) are also considered in this study.

In the atmospheric forcing dataset, we perturb the horizontal 10 m wind velocities, the downward longwave radiation, and the snowfall rate that are considered to be dominant factors for a winter period. In the ocean forcing dataset, the SSS and SST are perturbed since they are known to impact SIC and SIT through thermodynamic processes. Using the same perturbation system as TOPAZ4 (Sakov et al., 2012), the perturbations are spatio-temporally correlated with a decorrelation time scale of 2 days and a horizontal decorrelation length scale of 250 km, with the random field generator introduced by Evensen (2003). The random perturbation fields have standard deviations of $\sqrt{3}$ m/s for wind speed, $\sqrt{50}$ W/m$^2$ for the longwave radiation, 100% for snowfall rate (using an unbiased lognormal distribution), 0.1 °C for SST and 1 PSU (Practical Salinity Unit) for



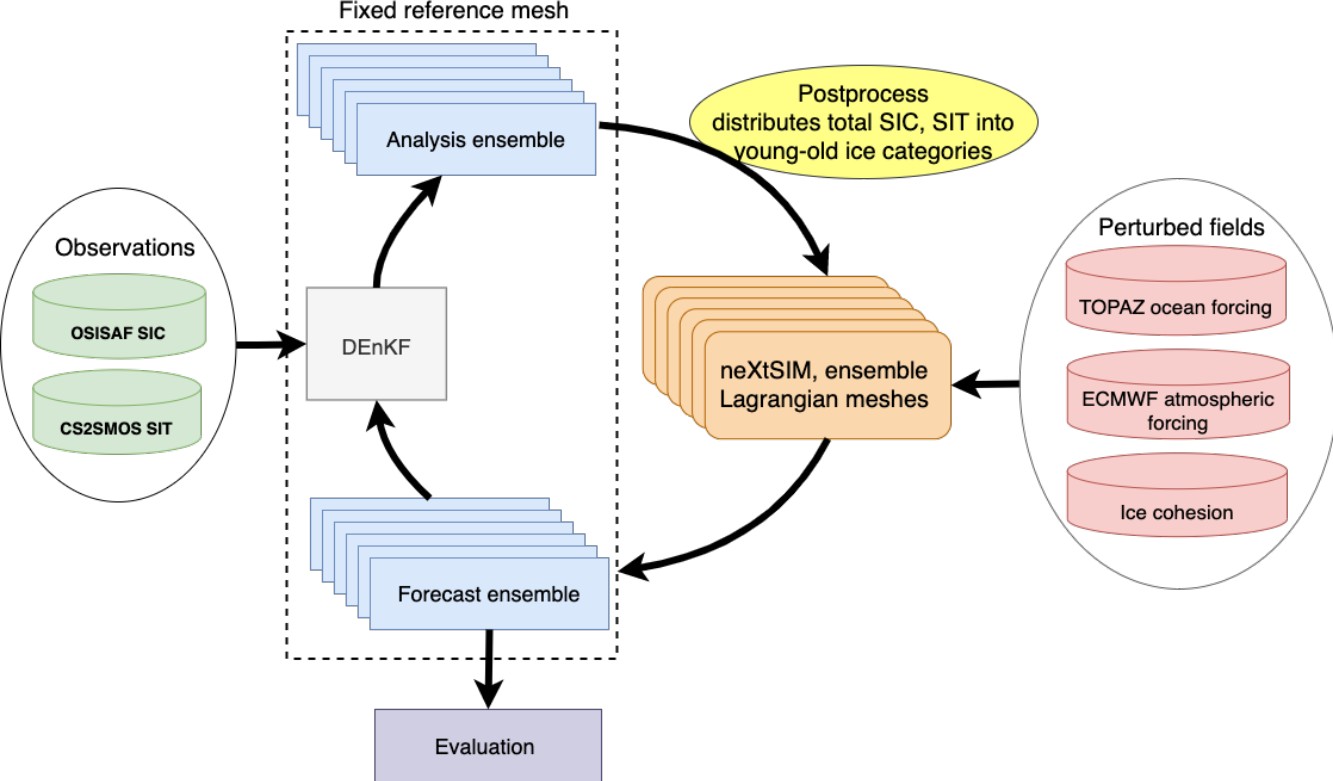

**Figure 1.** Flow chart of the ensemble-DA platform. DEnKF: analysis step; neXtSIM: model forecast step; OSI-SAF SIC/CS2SMOS SIT observation: assimilated observations; TOPAZ forecasts and ECMWF forecasts: upstream forcing data for neXtSIM; Field perturbations: see Section 4.1; Fixed Reference Mesh: forecast and analysis state variables are exchanged on a pre-defined regular static grid.

the SSS. To limit the impact of SST on the ice formation around the ice edge, we multiply the SST perturbations by the open water fraction in a grid cell. Following Sakov et al. (2012), the perturbations of horizontal wind velocities are derived from a surface-level pressure perturbation with a non-divergent constraint. The non-divergent constraint prevents the sea ice cover from breaking up excessively.

In addition to perturbing the upstream forcing dataset, the ice cohesion field is initialized as a heterogeneous random field following a uniform distribution between 20 and 40 kPa without spatial correlation (Cheng et al., 2020), which is different for each ensemble member. The cohesion field is kept constant unless a remeshing occurs, in which case the cohesion on the newly created model grid cells is given as the average of their nearest neighbors.

**4.2   Observation uncertainties**

The OSI-SAF SIC product provides an estimate of the instrument uncertainty to which we must add the related representation error (Janjić et al., 2018). The latter is notably challenging to evaluate and is usually computed based on pragmatic choices.





Following Sakov et al. (2012), we estimate the total observation error variance (sensor and representation errors) $\sigma^2_{\text{obs,SIC}}$ as a function of SIC,

$$\sigma^2_{\text{obs,SIC}} = 0.01 + (0.5 - |0.5 - c|)^2, \tag{1}$$

with $c$ being the SIC. A visual comparison (not shown) between the error variances from Eq. (1) and those specified in OSI-SAF SIC confirms that they agree qualitatively but that Eq. (1) gives slightly higher uncertainties in certain areas as aimed for. Due to the difficulties of retrieving SIT from remote sensing, the uncertainty of the CS2SMOS SIT product is relatively high. The uncertainty contains the sensor errors, the inverse model representation error, diagnosed observation errors, and the bias (inconsistency) between the CryoSat-2 and SMOS observations. Instead of using the CS2SMOS built-in uncertainty, we define the observation error variance $\sigma^2_{\text{obs,SIT}}$, referring to the empirical formula of Xie et al. (2018), as an increasing function of ice thickness $h_{ice}$,

$$\sigma^2_{\text{obs,SIT}} = \begin{cases} \min(0.2, 0.02e^{1.8(h_{ice}-3)}), & h_{ice} > 3m, \\ \max(0.02, 0.1e^{1.5h_{ice}}), & \text{otherwise.} \end{cases} \tag{2}$$

The coefficients in Eq. (2) are fine-tuned based on a 3-year-long dataset of observations.

## 4.3 Multivariate analysis update

Previous studies have shown that multivariate DA can improve the SIT even when the SIC alone is assimilated (Massonnet et al., 2015; Zhang et al., 2018; Mu et al., 2020). Hence, in this study, the SIT and SIC are included among the updated state variables in the analysis steps. At least one state variable (either SIC or SIT) in our state vector has observations in the following DA experiments. Moreover, because the freezing and melting of sea ice are strongly driven by the ocean boundary conditions, we also include the SSS and SST among the analyzed variables. Recalling that SSS and SST are prognostic variables in the slab ocean of neXtSIM. This implies that the DEnKF shall adjust the ocean state variables following the change of sea ice via the cross-covariances of these analyzed variables, although in view of the simplicity of the slab ocean model, this does not constitute a truly coupled data assimilation setup, see Penny and Hamill (2017). To summarise, the state vector for DA (i.e., the quantities to be updated at analysis steps) are absolute SIT, SIC, SST, and SSS.

Recall that neXtSIM includes the so-called effective SIT instead of the absolute SIT among its prognostic variables. To assimilate the observed absolute SIT, we would need to modify the observation operator to include the nonlinear mapping from the effective SIT to the absolute SIT. Instead, we opted to transform the effective SIT in the model output into the absolute SIT before applying assimilation. This choice allows for maintaining a linear observation operator in the DA algorithm.

As mentioned in Section 2.1, the model has three ice categories: open water, young ice, and old ice. Instead of using a multi-category update as in Massonnet et al. (2015), SIC and SIT in our state vector are both a sum of young and old ice, and we redistribute the analysis updates in the respective categories via a postprocess whose details are given in Section 4.5.3.



## 4.4 Mapping model states on the fixed reference mesh

At the analysis steps, the state vector indicated in Section 4.3 is mapped from the individual Lagrangian mesh of the members onto a fixed reference mesh. The fixed reference mesh has a horizontal resolution of about 12 km over the Arctic Ocean,

extracted from the global tripolar grid with a 0.25° resolution, referred to as ORCA-R025 (Bernard et al., 2006). The mapping is accomplished by a bi-linear interpolation. The analysis state vector is then interpolated back to the corresponding Lagrangian mesh of the ensemble members for the subsequent forecast step following Aydoğdu et al. (2019). Once the updated values of the physical quantities in the state vectors are projected back on each member's mesh, they evolve with the forecast run in which the Lagrangian meshes move according to the ice velocity fields.

## 4.5 The Deterministic Ensemble Kalman filter (DEnKF)

We use the DEnKF (Sakov and Oke, 2008) to assimilate the SIT and SIC observations. Unlike the stochastic EnKF, the DEnKF does not require the observations to be perturbed to avoid filter divergence. DEnKF is also conceptually more straightforward compared to other transform-based deterministic EnKFs (see Carrassi et al., 2018, for the differences between various EnKF flavors). The open-source DA software, EnKF-C version 2.8.0 (Sakov, 2014) is adopted to perform the DA. The EnKF-C

provides the functionality for automated quality control and ensemble inflation and localization, which are briefly recalled below.

### 4.5.1 Preprocessing and quality control

EnKF-C provides a toolbox to preprocess and perform observation quality control. Observations within our model domain are selected and interpolated onto the fixed reference mesh (see 4.4) by its build-in operator. Additionally, observations within

50 km of the coastline are removed to avoid spurious effects known to be present in these satellite-derived products near the coasts.

    When the forecast and observation uncertainties do not overlap (i.e., innovations larger than expected by the EnKF), the assimilation responds by giving large analysis increments, which may cause model imbalances in the subsequent forecast run. To reduce the impact of those large innovations, EnKF-C uses an adaptive quality control method introduced by Sakov

and Sandery (2017), and that artificially inflates the observation variance by an adjustable K-factor without removing the observations. The effect of this adaptive quality control method has been studied in Sandery et al. (2020, Figure 1) by comparing the original and modified observation error spread. In the case of severe model biases, K can be set to 1. While in our case, as demonstrated by the free run in Williams et al. (2021), we do not expect severe bias in neXtSIM and found that the default value K=2 maintains adequate analysis ensemble spread.

### 4.5.2 Inflation and localization

Idealized studies by Zhang et al. (2018) show that inflation and localization are necessary to improve sea ice forecast in the DA cycles. DEnKF already includes inflation of the analysis ensemble implicitly (Sakov and Oke, 2008). Nevertheless, inflating




the observation variance can further reduce the observations' effect and thus maintain the spread from the ensemble forecast. We thus intentionally multiply the observation variance for all observations by a factor of 2, but we do not use any other multiplicative inflation. Following Sakov et al. (2012), we use a localization radius of 300 km for sampling both SIC and SIT observations. This corresponds to an effective $e^{\frac{1}{2}}$-folding radius of about 90 km.

### 4.5.3 Postprocessing of nonphysical analyses

As stated in Section 4.3, we simultaneously update the SIC, SIT, SST, and SSS by the DEnKF. The important, albeit generally nonlinear, ocean and sea-ice interactions might not be fully captured by the error covariances that in turn determine the DEnKF updates, resulting in nonphysical model states. To mitigate this potential issue, we apply a sequence of postprocessing steps as follows. Firstly, as a sanity check, the ocean variables are capped by the SSS between 5 and 41 PSU, and the SST between the freezing point ($-0.057 \times$ SSS) and 35 °C, far out of reasonable values. Secondly, the postprocessing handles the analysis update of total SIC. To remove the effect of unreliable observations, SIC is set to zero wherever the analysis SIC $<= 15\%$ and SIC is capped by 100%. We assume the model uncertainty of SIC arises mainly from the young ice. Hence, the SIC increment, which is the difference between the analysis and the background total SIC, is firstly applied to update the young ice. The SIC of young ice is capped by 100% in case of a positive increment. If the young ice is completely removed by a negative increment, the rest of the negative increment is removed from the old ice. If SIC was removed altogether, SIT, ridge ratio, and snow depth are all set to 0 for no ice situation.

After processing the SIC increment, we make a selective update for SIT. In the experiments where SIT is assimilated, the SIT in the entire domain is updated by the analysis. On the other hand, if only the SIC is assimilated, the SIT updated by multivariate covariances is applied in a limited region where the analysis SIC $<= 90\%$. This is done to remove possible unrealistic multivariate updates of SIT in the pack ice; high SIC values reaching the upper bound of 100% do not fit the Gaussian assumption and sometimes lead to problematic bivariate error covariances with SIT. Our selection relieves the problematic updates in the DEnKF but retains the multivariate assimilation for intermediate SIC values. Then, the SIT of young ice and old ice is redistributed proportionally to the analysis fractions of young and old SIC. For "new" ice added by data assimilation, we assign 0.25 m SIT to the young ice and keep old ice at zero. If the updated SIT of either young ice or old ice falls thinner than 0.01 m, we remove that category by setting the relevant ice quantities to zero.

These updated sea-ice quantities then go through a final consistency check to be used in the model run. Especially, the internal ice temperature is updated based on the updated SIT, SIC, snow thickness, and water freezing temperature. Internal ice temperature must be at freezing point for regions without old ice. For open water areas without any ice, the consistency check ensures the SIT, SIC, and snow depth are all zero for both ice categories.

## 5 Experiment setup

We carry out four ensemble-DA experiments to investigate the impact of different observation products and assimilation frequencies. Details are summarised in Table 1 together with a free ensemble run for reference.





**Table 1.** Summary of the ensemble experiments. For all the experiments, the simulation period is from 18 October 2019 to 16 April 2020, and the ensemble size is 40.

| Exp. Name | Assimilation frequency, day(s) | | Postprocessing |
|-----------|------------|-------------|----------------|
|           | OSI-SAF SIC | CS2SMOS SIT |                |
| free run  | -          | -           |                |
| *SIC7*    | 7          | -           | Limited update SIT* |
| *SIT7*    | -          | 7           |                |
| *SIC7-SIT7* | 7        | 7           |                |
| *SIC1-SIT7* | 1        | 7           | Limited update SIT* |

\* When only SIC is assimilated, SIT in the model is updated with analysis SIT only if the
analysis SIC $<= 90\%$, see Section4.5.3.

In particular, in two of the experiments, hereafter referred to as *SIC7* and *SIT7*, we assimilate observations of only one quantity, either SIC or SIT, respectively, where the last digit indicates the weekly DA frequency. These experiments are intended to reveal the effect of multivariate assimilation on the observed and non-observed sea ice variables. Similar to previous studies (see, e.g., Xie et al., 2018), we investigate the joint assimilation of both SIC and SIT in two further experiments. In the experiment *SIC7-SIT7*, we assimilate SIC and SIT simultaneously on a weekly cycle. Recalling that the SST and SSS in

the slab-ocean component of neXtSIM are nudged towards the daily ocean surface forcing of TOPAZ4 with the relaxation timescales set as 5 days (see Section 2.2), the SST and SSS updated by weekly multivariate analysis are gradually "overwritten" in neXtSIM by the TOPAZ4 values. Hence, we increase the assimilation frequency and conduct the additional experiment *SIC1-SIT7*, whereby SIC is assimilated daily and SIT is assimilated weekly, following the satellite data availability. In this experiment, the effect of ocean nudging should be mitigated, and the impact of SIC observations intensified. Moreover, the

higher observational frequency may damp the non-linearity in the error evolution in subsequent analyses, potentially avoiding the violation of the Gaussian assumption at the basis of the DEnKF (see Bocquet and Carrassi, 2017, for a discussion on the effect of observation frequency and accuracy on the error evolution). Naturally, the downside of high-frequency assimilation is the increased computational cost (Lange and Craig, 2014; He et al., 2020). More frequent assimilation of SIT was not considered in view of the limited daily coverage of CryoSat-2 data in the ice pack. As a reference, we produce a free ensemble

experiment without DA constraints, referred to as free run hereafter.

     Because the CS2SMOS product is only available in the ice freezing season, all the experiments are conducted from 18 October 2019 to 16 April 2020 (182 days). The initial conditions of the experiments are generated from an ensemble spin-up run from 3 September to 17 October 2019 (45 days). The spin-up run is initialized from the neXtSIM-F forecasts (Williams et al., 2021) and integrated with different perturbations for each ensemble member as described in Section 4.1. All the DA

experiments and the spin-up run use 40 ensemble members. The ensemble size is chosen after evaluating the spread of a larger free run ensemble of 100 members during the spin-up period, noticing that the spread saturates for ensemble sizes above 40.



# 6 Results

In this section, we present the results from the model forecasts of the experiments in Table 1 from 18 October 2019 to 16 April 2020, validated by the observations before the sea-ice variables are assimilated. Although assimilated into the experiments, the observations of SIC and SIT are thus quasi-independent from the model forecasts because the observed values have not yet been assimilated.

Truly independent validation is also performed using the SID observations. The Section is structured to show results on different quantities: SIC and SIE in Section 6.2), SIT in Section 6.3, and SID in Section 6.4.

## 6.1 Definition of Metrics and Overall Performances

We use several metrics to validate the ensemble forecasts against the observations, thus evaluating the forecast skills of our system. All metrics are defined in observation space, i.e., in the observation grid onto which the model states are interpolated.

Bias and Root Mean Square Difference (RMSD) between the ensemble mean and the observations is utilized to assess the forecast skill. For a scalar variable field, such as SIC and SIT, the ensemble mean vector at time $t$ is defined as $\mathbf{x}(t) = \frac{1}{n}\mathbf{E}(t)\mathbf{1}$, where $\mathbf{x}(t) \in \mathbb{R}^m$, $m$ is the vector length, $n$ is the ensemble size, $\mathbf{E}(t) \in \mathbb{R}^{m \times n}$ is a matrix containing the ensemble members along its columns and $\mathbf{1} = (1,...,1)^{\mathrm{T}} \in \mathbb{R}^n$. The bias is - contrary to the innovations - defined as model-minus-observations $\mathbf{d}(t) = \mathcal{H}(\mathbf{x}(t)) - \mathbf{y}(t)$, where $\mathbf{y}(t) \in \mathbb{R}^o$ is the observation vector, $\mathcal{H}$ is the observation operator mapping from model states with two sea ice categories to the observations of total SIC and total absolute SIT on the observation grid. In our analysis we define a "robust" bias by removing grid points outside of the $[2\%, 98\%]$ quantiles of bias on the domain. This make $\mathbf{d} \in \mathbb{R}^{o'}$ with $o' \leq o$. The RMSD is defined as $\|\mathbf{d}(t)\|$, where $\|\cdot\|$ is the Euclidean norm also with outliers removed. We also compute the temporal average of the bias over the time period $[t_1, t_2]$, which is $\frac{1}{t_2-t_1}\int_{t_1}^{t_2}\mathbf{d}(t)\mathrm{d}t$.

For vector quantities such as the SID, the vector of ensemble mean at time $t$ is defined as $\mathbf{x}(t) = (\mathbf{x}_1(t), \mathbf{x}_2(t))^{\mathrm{T}} = \frac{1}{n}\mathbf{E}(t)\mathbf{1}$, where $\mathbf{x}_1(t), \mathbf{x}_2(t) \in \mathbb{R}^m$ represent the two horizontal orthogonal components, respectively, $\mathbf{E}(t) \in \mathbb{R}^{2m \times n}$ is the ensemble matrix with the first $m$ rows representing the first orthogonal component of the quantity, and the rest $m$ rows represent the second orthogonal component, and $\mathbf{1} = (1,...,1)^{\mathrm{T}} \in \mathbb{R}^{2m}$. The relevant observation vector $\mathbf{y}(t) = (\mathbf{y}_1(t), \mathbf{y}_2(t))^{\mathrm{T}} \in \mathbb{R}^{2o}$ has $o$ number of observations with its horizontal orthogonal components $\mathbf{y}_1(t), \mathbf{y}_2(t) \in \mathbb{R}^o$. The bias can be defined as the error of the magnitude of speed: $\mathbf{d}(t) = |\mathcal{H}(\mathbf{x}(t))| - |\mathbf{y}(t)|$, then RMSD is the error in speed $\|\mathbf{d}\|$, where $|\cdot|$ is defined as $|\mathbf{z}| = \sqrt{\sum_{i=1}^{2}\mathbf{z}_i \circ \mathbf{z}_i}$ where $\circ$ is a component-wise product, with $\sqrt{\ }$ being vector component-wise square root. In the vector case, the outliers are not removed. We also compute the Vector RMSD (VRMSD), $\|\mathcal{H}(\mathbf{x}(t)) - \mathbf{y}(t)\|$, which accounts for errors in speed and direction. Note that the metrics defined above follow Williams et al. (2021) with adjustments for ensembles.

We evaluate the SIE by adopting the integrated ice-edge error (IIEE, Goessling et al., 2016) and the spatial probability score (SPS, Goessling and Jung, 2018). The IIEE indicates the mismatch of SIE between the model outputs and observations usually occurring around the sea ice edge, which is designed to evaluate deterministic forecasts. Moreover, the IIEE can easily be decomposed into overestimation and underestimation of SIE. Using the ensemble mean, we compute the IIEE over- and under-estimated components, $O$ and $U$. The SPS is an extension of the IIEE for ensemble forecasts. However, the SPS does





**Table 2.** Summary of the metrics for SIC, SIT, and SID: spatio-temporal averaged bias, RMSD, and VRMSD among the experiments. SIC is validated against OSI-SAF SIC, SIT is validated against CS2SMOS SIT, and SID is validated against OSI-SAF SID. The best value of each ice feature among the experiments is bolded.

| Exp. Name | SIC | | SIT (m) | | SID (km/2days) | | |
|---|---|---|---|---|---|---|---|
| | Bias | RMSD | Bias | RMSD | Bias | RMSD | VRMSD |
| free run | **0.014** | 0.032 | 0.168 | 0.304 | 1.696 | 4.224 | 8.265 |
| *SIC7* | 0.015 | 0.031 | 0.123 | 0.247 | 1.680 | **4.210** | 8.254 |
| *SIT7* | 0.017 | 0.034 | **-0.051** | **0.118** | 1.400 | 4.944 | **7.099** |
| *SIC7-SIT7* | 0.017 | 0.033 | -0.090 | 0.155 | **1.279** | 4.935 | 7.130 |
| *SIC1-SIT7* | 0.015 | **0.027** | -0.113 | 0.173 | 1.296 | 4.925 | 7.115 |

not provide information on overestimation or underestimation. Hence, we utilize both metrics in the SIE assessment. Formulas of the relevant metrics can be written as:

$$O = \int_A \max\left(c\left(\overline{\mathcal{H}(\mathrm{SIC}_f)}\right) - c(\mathrm{SIC_o}), 0\right) dA, \quad U = \int_A \max\left(c(\mathrm{SIC_o}) - c\left(\overline{\mathcal{H}(\mathrm{SIC}_f)}\right), 0\right) dA, \quad \mathrm{IIEE} = O + U, \quad (3)$$

where $A$ is the area of model domain, $c(x)$ is a sea ice indicator function such that $c(x) = 1$ for $x >15\%$ otherwise $c(x) = 0$, where $x$ represents ice concentration; the subscripts $o$ and $f$ stand for the observed and forecast concentrations, respectively,

while $\overline{\mathcal{H}(\mathrm{SIC}_f)}$ indicates the ensemble mean of the overall SIC forecast in the observation space. Note that only the 15% concentration isoline is used by the IIEE and a large share of the concentration information is not taken into account. Finally,

$$\mathrm{SPS} = \int_A \left\{P[c(\mathcal{H}(\mathrm{SIC}_f))] - c(\mathrm{SIC_o})\right\}^2 dA, \quad (4)$$

where $P$ is the ensemble probability of SIE of each model grid point.

Before discussing the skills in each of the considered variables in detail, we summarize their performances by providing

the grid-weighted spatial and temporal averages of the bias, RMSD, and VRMSD in Table 2. The best result per variable and metric is highlighted in bold.

### 6.2  Evaluation of sea ice concentration and extent

Figure 2(a) shows the temporal evolution of the area-weighted average of the ensemble mean SIC from the experiments in Table 1. The black curve indicates the OSI-SAF SIC observations. All experiments capture well the observed increase in SIC.

Nevertheless, although the free run is almost indistinguishable from all the DA experiments, it shows a slight overestimation from November onwards. Among the DA runs, the experiment *SIC1-SIT7* (dashed purple line) shows a marginally better agreement with the observations than the others.

Figure 2(b) and (c) display the time series of bias and RMSD of SIC, respectively. Overall, all the experiments show excess sea ice with minor differences. The SIC bias is generally smaller when SIC is assimilated, as expected. The experiment with





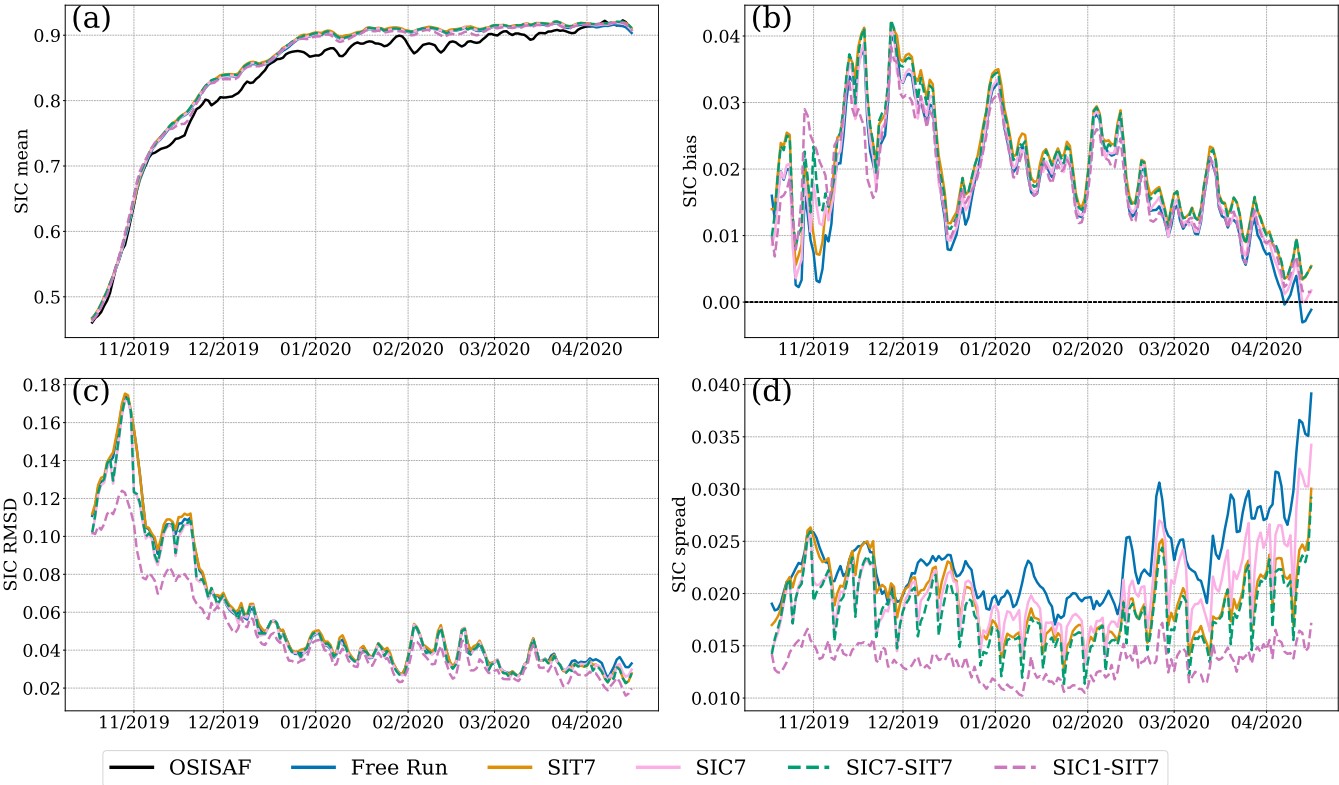

**Figure 2.** Time series of spatially averaged SIC from the model forecast against OSI-SAF SIC from 18 October 2019 to 16 April 2020. a) ensemble mean SIC; b) bias; c) RMSD; d) ensemble spread (standard deviation).

daily SIC observations, *SIC1-SIT7*, generally gives the smallest biases. We see, however, that the free run has the smallest bias in the first weeks, occasionally in December 2019 and April 2020. The free run shows also the (slightly) smallest average bias of $0.014$ (see Table 2). This undesired, albeit small, increase of bias in the DA experiments is due to the asymmetric effect of DA in the cases of local overestimation and underestimation. We shall clarify this further when discussing Figure 5 later.

In contrast to the bias, we see an obvious impact of DA on the RMSD (panel (c)). In particular, *SIC1-SIT7* shows the lowest
RMSD consistently throughout the whole period (average value of $0.027$ in Table 2). This is a notable reduction compared to the other experiments, which show very similar values. The RMSD decreases mostly during fall 2019 (a relative reduction of up to $27\%$) and then oscillates during spring 2020. The effect of DA becomes smaller after January 2020 when the sea ice covers the majority of the Arctic Ocean (see the increase of SIC in Figure 2(a)), and there is not much room for action.

The DEnKF estimates the uncertainty via the ensemble spread, a flow-dependent proxy of the forecast error standard devi-
ation. Comparing the spread of ensemble-DA experiments with the free run provides an estimate of how much uncertainty is reduced by the DA. We show the time series of the ensemble spread from all the experiments in Figure 2(d). All experiments maintain a stable spread except for a seasonal increase in April 2020 (expected from Lisæter et al. (2003)), and the assimilation



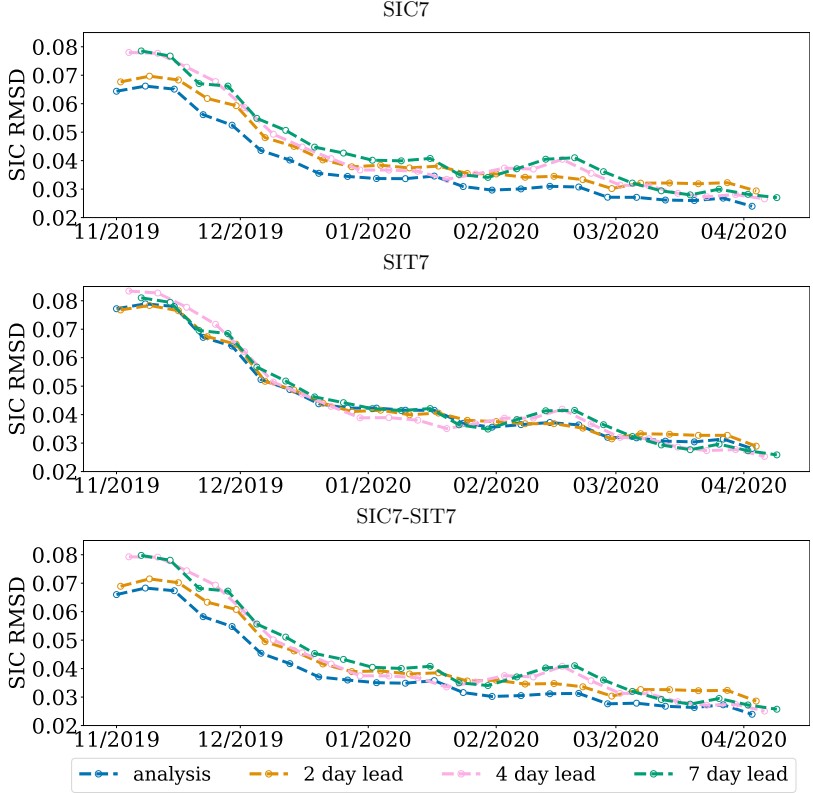

**Figure 3.** SIC RMSD of four weeks moving averaged forecast regarding the assimilation time. Each line represents a different leading time.

reduces the ensemble spread compared to the free run, although to a different degree depending on the experiment. In the experiments with weekly assimilation of either SIT or SIC, the spread decreases significantly after 1-day forecasts from the

analysis. However, it then quickly builds up during the two following days of forecasts, reaching levels close to the free run from October 2019 to January 2020, showing that the assimilation of SIC has a short memory compared to the assimilation window. The weekly assimilation of SIC is slightly more efficient from April onwards, consistently with Lisæter et al. (2003). However, only the daily SIC assimilation experiment can efficiently reduce the SIC errors, *SIC1-SIT7*, showing a relatively smaller spread across the full period. Both spread and RMSD are reduced by about 0.005. Importantly, these results show that

thanks to the continuous perturbations applied to the external forcing, the ensemble system can maintain spread even in the winter when the system's internal variability is tiny.

To get a clearer picture of the improvements in the SIC forecast skill by weekly assimilation, we present the SIC RMSD from the perspective of the lead time in Figure 3. Each curve represents the temporal evolution of the moving average in a four-week window of the analysis and forecast error after 2, 4, and 7 days of lead time. The clustering and spread of the curves

at a specific time (viewing vertically) indicates the growing forecast error with the increased lead time: we observe that both *SIC7* (top row) and *SIC7-SIT7* (bottom row) show a clear growth of the RMSD with increasing lead time, while the forecast



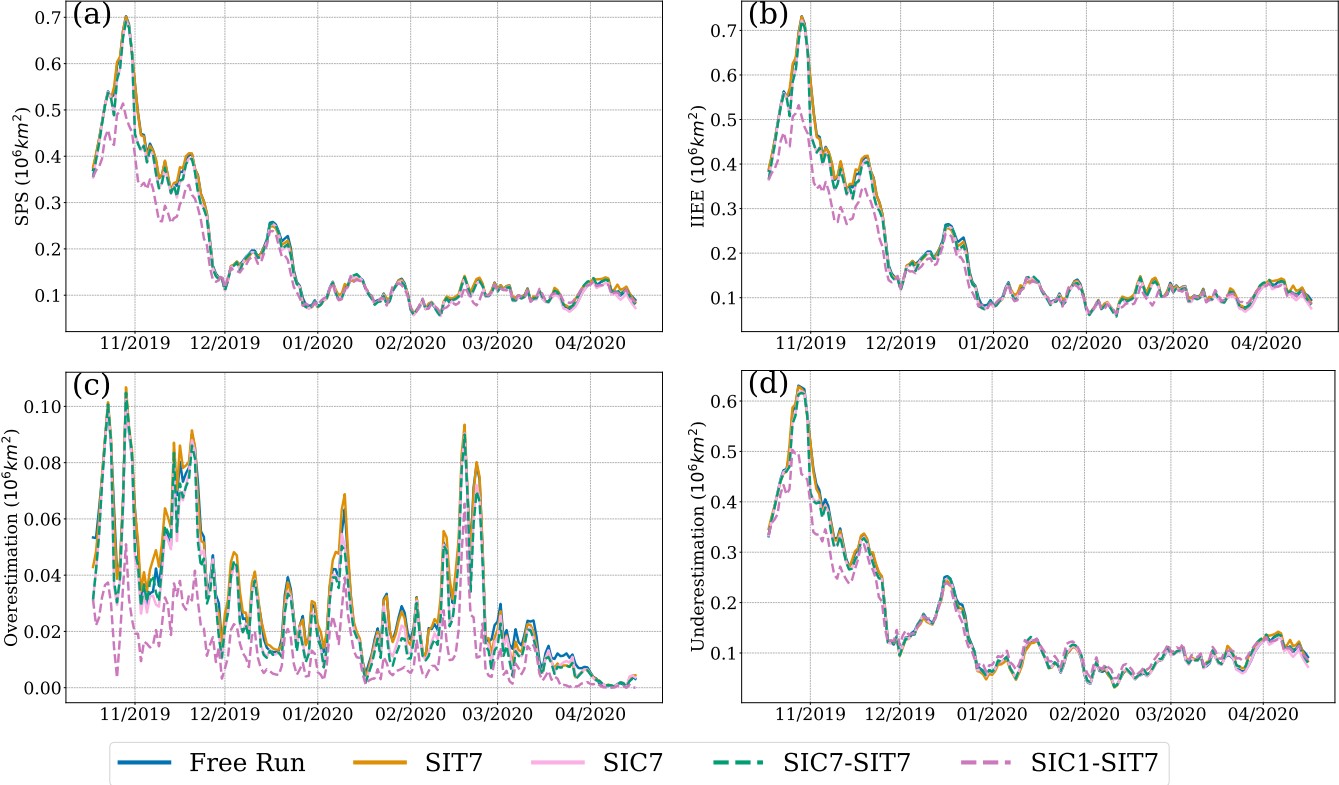

**Figure 4.** Metrics of sea ice extend validated against OSI-SAF SIC over time. (a) Spatial probability score (SPS) is calculated using the full ensemble forecasts; (b) Integrated ice-edge error (IIEE) is calculated using the ensemble means and split between its over- and underestimation components in panels(c)(d), respectively.

errors are independent of lead time in *SIT7* (middle row). This is a result of the reduction of model error in the analysis (after lead day 7) in *SIC7* and *SIC7-SIT7* with increasing error in the subsequent model forecast (in lead day 1), which is not reflected in the daily-averaged time series in Figure 2 above. We observe that the errors saturate in a range of two to seven forecast lead

days depending on periods, which we attribute to the variability of weather conditions. This highlights the improvements in the analysis due to the assimilation of SIC but also reveals that the multivariate assimilation of SIT has little influence on SIC. With the decreasing SIC RMSD over time (viewing horizontally), the forecast error among different lead times gets similar to each other, showing the reduced effect of the assimilation on the forecast in the winter season.

The SIE is also evaluated against the OSI-SAF SIC product from a domain-integrated view of the SIC. Figure 4 shows the

time series of the (a) SPS and (b) IIEE, (c) overestimated, and (d) underestimated components of IIEE (see Section 6.1). The SPS and IIEE are almost identical for all runs. This is typical of a "healthy" ensemble that the ensemble forecasts and their ensemble mean are statistically indistinguishable. The SPS and IIEE behave similarly to the SIC RMSD (see Figure 2(c)), which decreases to a steady state around January 2020 with small and rapid variations afterward. The overestimated and





underestimated components of IEEE based on the ensemble mean give a different view of the bias of SIE in the ensemble-mean forecasts: the SIE is primarily underestimated comparing Figure 4(c) and (d), which seems counter-intuitive when the biases were showing a general overestimation of SIC. The discrepancy is likely caused by local underestimated SIC in the Beaufort Sea versus general overestimation over the Arctic, which will be further reviewed in Figure 5.

The improvement of SIE due to DA is most visible in *SIC1-SIT7* compared to the others, highlighting the importance of frequent SIC assimilation. The underestimated component of the IIEE is improved in the first months only, consistently with the impossibility of adding more ice in a fully packed Arctic. In contrast, the overestimation is improved throughout the whole period due to the more frequent (daily) assimilation of SIC. In absolute IIEE numbers, reducing the underestimation makes most of the impact of assimilation, while in relative terms, the overestimation may seem easier to correct with daily assimilation.

To further demonstrate the effect of DA, Figure 5 shows the spatial distribution of the monthly SIC average bias in selected months, representative of the SIC evolution. The top row displays the observed SIC, and the SIC biases from the different experiments are then shown below. The distribution of SIC bias is similar in all experiments showing an overall positive bias in the ice pack where concentrations are high (>80%). High bias regions are also located near the marginal ice zone, in coastal seas, and east of Greenland. The SIC bias in the marginal seas is visibly reduced in the *SIC1-SIT7* compared to the free run and the other DA experiments, among which there are no noticeable differences. These results agree with the time series of bias and RMSD of SIC and SIE shown in Figures 2 and 4.

The figure offers a resolution of the apparent contradiction between the overestimated SIC bias and the underestimation-dominated IIEE decomposition: all experiments generally predict smaller SIE (green curves) compared to the observation (red curves) near the open water boundaries. This then causes the predominant underestimation of SIE seen in Figure 4, but within the 15% isoline, most of the biases are positive, summing up to a positive SIC bias in Figure 2. In November 2019 (left column), the large bias in the Beaufort and Chukchi Seas and the Kara Sea contributed to most of the SIC RMSD seen in Figure 2. As the sea ice freezes gradually, the SIC bias disappears as the ice expands to the Bering Strait and the Siberian coasts. Meanwhile, the SIE bias decreases over time, as the SPS/IIEE indicated above. Disagreements between modeled and observed SIC and SIE remain in the Nordic Seas: the Greenland Sea, the Barents Sea, and the Kara Sea in January and March 2020, which are only slightly affected by the assimilation and likely biased towards their ocean boundary condition. The delayed ice growth in the model is mainly related to the warm ocean condition from TOPAZ4 since the atmosphere temperature is commonly below the freezing point. Because the SIC and SIE evolution primarily depends on the underneath ocean boundary condition, we argue that during the model forecast, the daily TOPAZ4 data counteract the effect of weekly assimilation on the ocean states. This is not the case in *SIC1-SIT7*, which updates daily the analysis SST and SSS in the slab ocean via DA. This argument is in agreement with Williams et al. (2021) which blamed the model for the underestimation as the lack of input of detached land-fast ice and fast melting in the seas. This phenomenon is further discussed in Section 7.





**Figure 5.** Monthly average of SIC observations (first row) and SIC bias between the model forecasts and the OSI-SAF SIC product. Panels from left to right are November 2019, January 2020, and March 2020. Red and green lines are the sea ice edges (applying the classical threshold of 15% SIC) from the observations, and the ensemble mean forecasts.





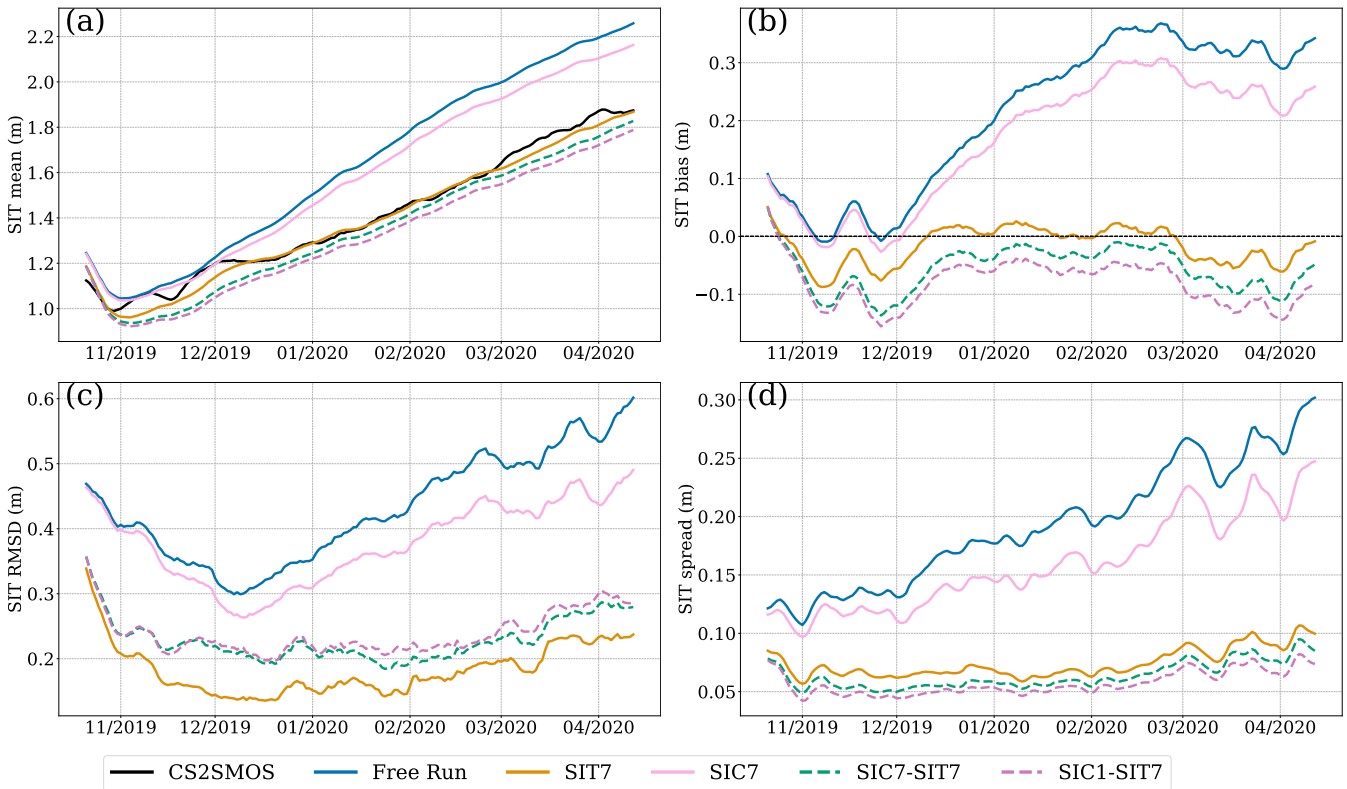

**Figure 6.** Time series of spatial-averaged SIT from the model forecast against CS2SMOS from 18 October 2019 to 16 April 2020. a) ensemble mean SIT; b) bias; c) RMSD; d) ensemble spread (standard deviation).

### 6.3 Evaluation of sea ice thickness

We now examine the SIT forecast skill of our experiments. Similar to Figure 2, Figure 6 shows the (a) spatial mean, (b) bias, (c) RMSD, and (d) ensemble spread of SIT validated against the CS2SMOS SIT product. We can see that all experiments capture the mean SIT evolution in the measurements. In particular, they track the SIT increase after a short period of decrease in

October 2019. Nevertheless, the free run and *SIC7* significantly overestimate the mean SIT (see also panel (b)) from December 2019. It is worth mentioning that, thanks to the multivariate assimilation of SIC, the experiment *SIC7* is slightly better than the free run. As expected, all the experiments in which SIT is assimilated show, in general, much better skill in predicting SIT. Nevertheless, we also observe a counter-intuitive impact of assimilating both SIC and SIT (i.e., experiments *SIC1-SIT7* and *SIC7-SIT7*) yielding poorer SIT than when assimilating SIT alone. The differences among the DA experiments are more

evident in the bias (panel (b)) and the RMSD (panel (c)).

The free run shows the largest positive bias and RMSD of SIT over time. Errors are reduced by all DA experiments, with *SIC7* being the least effective; see also Table 2. The experiment *SIT7* gives the lowest bias (close to zero) and the smallest RMSD: the spatio-temporal averages for the bias and the RMSD are -0.051 m and 0.118 m respectively (see Table 2). *SIC7-*





*SIT7* and *SIC1-SIT7* show relatively larger RMSD and similarly larger negative bias than *SIT7*. We also note that assimilating

SIC daily (experiment *SIC1-SIT7*) gives both larger bias and RMSD than assimilating SIC weekly (experiment *SIC7-SIT7*). The time-averaged bias and RMSD (cf Table 2) are respectively -0.113 m and 0.173 m for *SIC1-SIT7* and -0.09 m and 0.155 m, for *SIC7-SIT7*. Although this degradation seems to contradict the positive impact of *SIC7* over the free run, we believe that the slight reduction in SIT prediction skill is caused by artefacts of the DEnKF update applied to SIT and SIC, whereas the relationship between the two variables is nonlinear. A practical remediation would be to attenuate the effect of SIC

observations on SIT analysis. We will discuss this in Section 7. In contrast to the similar SIC spread in Figure 2(d), the SIT spread in Figure 6(d) exhibits two groups. The top lines show a large increase of the SIT spreads in the experiment *SIC7* and the free run without SIT assimilation. The unconstrained free run shows the largest spread, which is slightly reduced in *SIC7*. In the lower lines, all the experiments assimilating SIT show much smaller spreads with a slower increase toward the end of the simulation period. *SIC1-SIT7* shows the lowest spread, slightly smaller than all other experiments. We attribute this behavior

to the lower variability of the SIT that is constrained by the assimilation of SIT data.

Figure 7 shows spatial maps of the monthly average of the SIT bias against CS2SMOS SIT data in November 2019, January 2020, and March 2020, from left to right. The first row panels show the observed SIT while the others present the bias of the different experiments. The time evolution of the bias pattern (from November 2019 to March 2020) depends on the different assimilation strategies. The free run has a negative SIT bias in November 2019 up to about 1 m mainly in the Beaufort and

Chukchi Seas, East Siberian Sea, Laptev, and Kara Seas. It also shows that the positive bias is predominant in the rest of the Arctic, especially in the Canadian Archipelago, the northern coast of Greenland (>1 m). In January and March 2020, the positive bias is predominant in the Arctic region except for the band of negative bias from the Nansen Basin and the Laptev Sea occurring in March 2020. SIT is overestimated near the coast all over the Arctic, indicating too much ice ridging in the model. *SIC7* shows slightly reduced SIT bias from the free run due to the multivariate assimilation of SIC, as discussed in Figure 6.

In line with the results in Figure 6, *SIT7* has the smallest bias. In particular, *SIT7* successfully reduces most of the large SIT bias in the Canadian Archipelago and Greenland, Barents, and Kara Seas found in the other experiments. The remaining SIT bias in *SIT7* appears concentrated near the coasts and ice edges. It is because of no DA applied within 50km coast zones and the high SIT uncertainty in the CS2SMOS data. By assimilating SIC and SIT observations, *SIC7-SIT7* shows a similar pattern as *SIT7* but a more pronounced negative bias. The bias patterns of *SIC1-SIT7* and *SIC7-SIT7* are similar overall, implying the

weak influence of assimilating SIC on SIT again.

## 6.4 Evaluation of sea ice drift

The drift of the sea ice, its directions, and speed, is important both for operational forecasts and atmosphere-ice-ocean interactions. The main physical mechanisms of sea ice drift are the wind drag at the sea ice surface and the internal sea-ice forces. We are thus interested in assessing how our DA-based predictions reproduce the observed SID: this is validated using the OSI-SAF

SID product, which is not assimilated in our experiments.

Figure 8 shows the (a) ensemble mean, (b) bias, (c) RMSD, and (d) VRMSD of the SID velocity as a function of time for the experiments. The free run can already reproduce the observed drift variations without DA. This makes it challenging for DA





**Figure 7.** Same as Figure 5 but for monthly averaged SIT bias between the ensemble runs and the CS2SMOS SIT product.

.





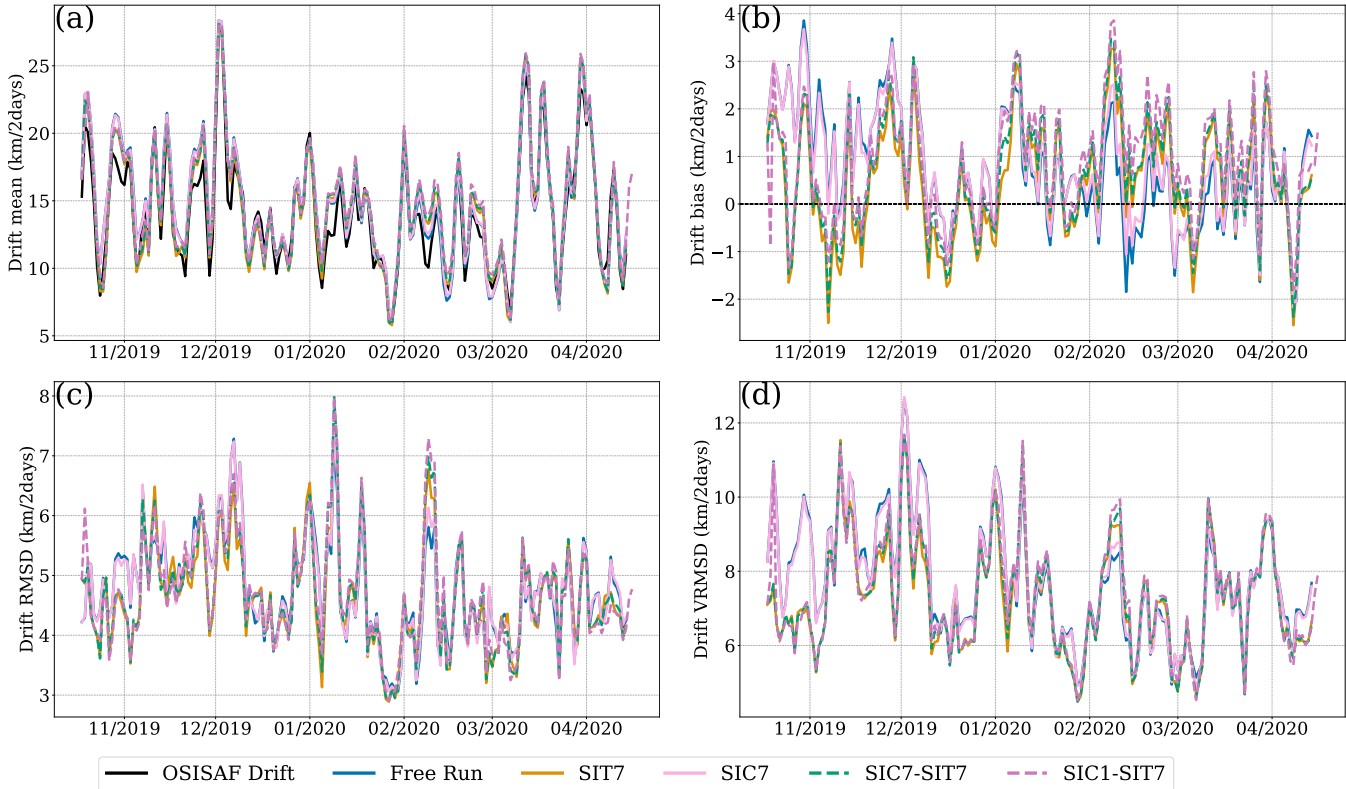

**Figure 8.** The time series of the (a) mean drift speed, (b) bias, (c) RMSD, and (d) vectorial RMSD against the OSI-SAF ice drift.

to bring visible improvements. However, some differences are noticeable in the bias and VRMSD from the time series shown in panels (b) and (d), respectively. For example, from October to December 2019, the *SIC7* experiment and the free run are

similar and show mostly positive drift bias. In contrast, *SIT7*, *SIC7-SIT7*, and *SIC1-SIT7* experiments are closer, with lower bias than the free run and *SIC7* and sometimes negative biases. In all cases, the drift bias varies between -2 to 3 $\mathrm{m/s}$ in panel (b). We interpret the results for RMSD and VRMSD along similar lines, in agreement with the spatio-temporal average given in Table 2. In particular, the bias in *SIC7-SIT7* is the lowest with 1.279 $\mathrm{km/2days}$. *SIT7* has the best VRMSD as 7.099 $\mathrm{km/2days}$, which is slightly smaller than *SIC7-SIT7* (7.13 $\mathrm{km/2days}$) and *SIC1-SIT7* (7.115 $\mathrm{km/2days}$). In contrast, the smallest RMSD

is from *SIC7* at 4.21 $\mathrm{km/2days}$ and the free run, outperforming experiments with SIT assimilation, but a close look at the time series shows that the SIT assimilation runs have the lowest errors most of the time, except for a short episode in February 2020, which is probably caused by random values of the perturbations and are not considered as representative. Overall, we observe small improvements in the modeled SID when we assimilate SIT.



## 7 Discussion and conclusions

We propose an ensemble-based data assimilation system for the Lagrangian sea ice model, neXtSIM, to enhance its Arctic sea ice forecast skill. The DEnKF is applied to work with a time-dependent, non-conservative model mesh, following the projected EnKF strategy introduced Aydoğdu et al. (2019). The Lagrangian nature of the model implies that each ensemble member evolves on an independent adaptive-moving mesh. At analysis times, we interpolate the ensemble members and the observations onto a fixed reference mesh, where the analysis is carried out. The analysis state variables are then projected back

to the individual member meshes. The ensemble is generated by perturbing internal model parameter and external forcings. Namely, the ice cohesion, the atmospheric forcing (10 m wind velocities, longwave downwelling radiation, and snowfall rate) from the ECMWF product, and the oceanographic forcing (SST, SSS) from the TOPAZ4 forecast. The system assimilates satellite-based sea ice products (CS2SMOS SIT and OSI-SAF SIC) and updates the state vector variables SIC, SIT, SSS, and SST by DEnKF. The model decomposes the sea ice into age categories, but the analysis is performed on the total ice. Therefore,

empirical postprocessing steps are introduced to redistribute the analysis of sea ice states into young and old categories and avoid imbalances caused by nonphysical updates.

Ensemble-DA experiments are carried out from October 2019 to April 2020 (Table 1) with different assimilation strategies. The results show that the forecast skill of sea ice improves and benefits from the assimilation. The forecast skills of SIC and SIE are the best in the *SIC1-SIT7* experiment with daily assimilation of SIC (Figures 2 and 4). The daily assimilation reduces

the model uncertainties, e.g., in the SIC RMSD (Figures 2) and spread (Figure 3). The underestimation of SIE is reduced in the Beaufort-Chukchi Seas and the Kara-Barents Sea in the *SIC1-SIT7* (Figure 5). The IIEE is roughly comparable to those obtained by the operational TOPAZ4 system, with a large part of the errors caused by underestimation (See the validation time series for sea ice in Copernicus Marine Service website (last accessed: 12 August 2022). The similarities of IIEE results are thus likely caused by areas of too warm ocean surface temperatures. Nevertheless, we also note that the IIEE may contradict the

SIC statistics since it only uses an isoline of the SIC. Moreover, we observe a large reduction of the (both positive and negative) model SIT biases by assimilating the CS2SMOS SIT data (Section 6.3). Specifically, the experiment *SIT7* cuts the model SIT bias almost to zero, and its corresponding RMSD is the lowest among all experiments. The SIT errors are also reduced in the joint SIC-SIT assimilation experiments, although to a lesser extent, which implies that the apparent benefit of multivariate DA becomes detrimental when assimilating both SIT and SIC, which is further discussed latter. Furthermore, small improvements

on SIT are visible when assimilating SIC only, *SIT7*, which agree with the small effects from multivariate DA found in the data assimilation framework of the coupled ocean-ice model, ROMS-CICE (Fritzner et al., 2019). Although the improvements on the SID are less apparent, the model performs on par with the earlier results with direct insertion (Williams et al., 2021) and outperforms the TOPAZ system using an older sea ice rheology and the DEnKF.

Three points in the results are further discussed as follows: Firstly, assimilating SIC in the joint SIC-SIT assimilation exper-

iments (*SIC1-SIC7* and *SIC7-SIT7*) causes more negative SIT bias than in the *SIT7*, which assimilates only SIT (see Figure 6 and 7). This undesired effect arguably comes from inadequate cross SIT-SIC error covariance in the DEnKF. The modeled SIC and SIT are generally positively correlated in the ensemble (not shown), especially when forming new ice at the ice edge.





Given that OSI-SAF SIC data are systematically lower than the model in most of the Arctic (see Figure 5), this generally positive SIC-SIT correlation implies that the analyzed thickness is also generally thinner after SIC assimilation. Consequently, the SIC assimilation tends to transfer the bias to the SIT. The influences of SIT-SIC cross-covariance get naturally stronger with more frequent DA in *SIC1-SIT7* compared with *SIC7-SIT7* (see Figure 6 and 7), leading to a stronger underestimation of SIT. This deterioration contradicts the case when SIC alone is assimilated and improves the SIT over the free run, but these improvements were most visible near the ice edge (see Figure 7) where the SIT-SIC covariance is more linear and because a correct location of the ice edge indirectly improves the thickness of newly formed ice.

The second point concerns a challenge inherent to ensemble-DA methods. The observed-unobserved variables in the sea ice are often non-linearly related. This limits the efficacy of the EnKF schemes, particularly for small ensemble size (e.g., Kimmritz et al., 2018). In our experiments, an issue is that both SIT, and SIC are bounded variables (positive and less than 1 for SIC) and thus clearly non-Gaussian. Furthermore, in virtue of the two sea ice categories in the model, the local relationship between the modeled SIT and SIC is often nonlinear. To mitigate the impact of wrongly computed analysis corrections, we intentionally limited the SIT update to regions where SIC < 90% (see Section 4.5.3).

Thirdly, the DA effectively reduces the uncertainty of the quantities directly observed; in practice, when, e.g., SIT or SIC is the sole observable. However, this benefit of DA is less pronounced when multi-variables are jointly assimilated. For example, assimilating SIT alone, *SIT7* cuts the SIT bias by 61% over the free run. In comparison, the improvement in the SIT bias by jointly assimilating SIC and SIT is smaller. Specifically, the bias reduces by 49% in *SIC7-SIT7* and 43% in *SIC1-SIC7* from the free run. This reduction has been a well-known effect since the early days of data assimilation in weather forecasting: when more terms are introduced in a cost function, the optimal solution becomes the best compromise and fits each term less closely individually. So in our case, the SIT and SIC observations compete with each other, and the joint assimilation performs somewhat poorer than the assimilation of the SIT and SIC individually.

This work suggests the feasibility of implementing an ensemble-based assimilation method for a Lagrangian sea ice model running on a moving/adaptive mesh and shows that the model benefits from data assimilation. It indicates a promising direction for the future development of the neXtSIM-F forecast system distributed through the EU Copernicus Marine Environmental Monitoring Service for the Arctic, using an ensemble-DA framework instead of the current deterministic data assimilation approach. We recommend that it is sufficient to assimilate the CS2SMOS SIT product weekly for the Arctic winter sea ice forecasts to significantly enhance the SIT forecast and slightly improve the SID forecast. However, the assimilation of SIC in a stand-alone model is limited by the accuracy of the ocean boundary conditions, and our attempt to include the slab ocean SST and SSS in the state vector has not been successful.

In future work, in implementing the presented method in the operational system, an evaluation will be carried out on multi-season and multi-year forecasts. Indeed, this first study focused on the winter season due to the limited availability of CS2SMOS products. The situation would be more complicated in the summer scenario since the sea ice dynamics are much more active. In particular, we do not expect the free run to be as skillful in predicting SIE as it is in the winter, see Williams et al. (2021), leaving more room for improvements thanks to DA, and the assimilation of SIC could be more effective. In agreement with the findings in Williams et al. (2021), the overall increase in performance on predicting SID when using DA





is insignificant; the free run already had an excellent fit to the observations. However, we expect this could be improved if assimilating drift observations directly or indirectly through ice deformations and including sea ice damage into the state vec-

tor. Additional modifications to the state vector and observation operator may include the multi-categorized sea ice properties instead of the total values. It would avoid the need for heuristic and empirical choices on redistributing SIC and SIT into young and old ice categories (Kimmritz et al., 2018). Moreover, the use of a variable-based localization is another potential venue for improvements. SIT very likely has a longer correlation length than SIC (Blanchard-Wrigglesworth and Bitz, 2014). Zhang et al. (2018) suggested a small localization to optimize the performance of SIC assimilation and a larger localization for assimilating

SIT based on a series of perfect model observing system simulation experiments with the version 5 of the CICE model and the EnKF.

The stand-alone sea-ice data assimilation system inherits warm biases from the ocean forcing, which limits the efficiency of SIC assimilation. This should be improved in a coupled ice-ocean model to be pursued in further work.

*Data availability.* The ECMWF atmospheric data is available on the ECMWF website (https://www.ecmwf.int/en/forecasts/datasets/set-i#

I-i-a_fc). The TOPAZ4 data is available from Copernicus Marine Environment Monitoring Service - The Arctic ocean physics analysis and forecast data (https://resources.marine.copernicus.eu/product-detail/ARCTIC_ANALYSIS_FORECAST_PHYS_002_001_a/INFORMATION). The OSI-SAF SIC (SSMIS) data is available at https://osi-saf.eumetsat.int/products/osi-401-b. The OSI-SAF SID data is available at https://osi-saf.eumetsat.int/products/osi-405-c.

*Author contributions.* SC conducted the experiments. SC, AA, and YC developed the codes, performed the data analysis, and wrote the

paper. All authors have contributed to the experiment's design, interpreting the results, the discussion, and writing the paper.

*Competing interests.* The authors declare that they have no conflict of interest.

*Acknowledgements.* We thank Dr. Pavel Sakov for helpful discussions and improvement regarding the EnKF-C code and Dr. Jiping Xie for contributing the TOPAZ interface to sea ice observations. We thank the EUMETSAT OSI-SAF centre for providing the sea ice concentration and SID data, https://osi-saf.eumetsat.int/. The production of the merged CryoSat2-SMOS sea ice thickness data was funded by the ESA

project SMOS & CryoSat-2 Sea Ice Data Product Processing and Dissemination Service, and data from 9/2019 to 4/2020 were obtained from AWI.

The work is funded by the DASIM-II grant from ONR (grant no. N00014-18-1-2493 and N00014-18-1-2204). AC, CJ, AA, and PR acknowledge the support of the project SASIP funded by Schmidt Futures – a philanthropic initiative that seeks to improve societal outcomes through the development of emerging science and technologies. SC and LB were co-funded by the FOCUS project from the Research Council

of Norway (grant no. 301450), and AC and YC are also supported by UK National Centre for Earth Observation (grant no. NCEO02004).

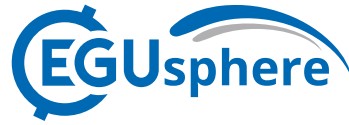

Computations were carried out on the Norwegian Supercomputing Infrastructure Sigma2 (grants nn2993k for computing and NS2993K for data storage).



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
