# Peer review of "Arctic sea ice data assimilation combining ensemble Kalman filter with a novel Lagrangian sea ice model for the winter 2019-2020"

_EGUsphere, 2022_

## Referee Comment (RC1)

Review of the manuscript „Novel Arctic sea ice data assimilation combining ensemble Kalman filter with a Lagrangian sea ice model" by Sukun Cheng et al.

The paper presents an experience of implementing a Kalman-type ensemble-based filter to combine sea ice concentration (SIC) and thickness (SIT) observational information with a Lagrangian sea ice model. In particular, the authors assimilate SIC from the Ocean and Sea Ice Satellite Application Facility (OSI-SAF) and the merged SIT product from CryoSat-2 and SMOS satellite missions into the Lagrangian sea ice model neXtSIM with the deterministic Ensemble Kalman filter (DEnKF). The filter analysis is performed on the ensemble of Lagrangian model states individually interpolated to a reference grid. The updated states are projected back onto the temporarily variable model mesh to reinitialize the model for the next forecast phase. The sea ice forecasting system is evaluated for the Arctic Ocean over the 2019/2020 winter time period. The OSI-SAF sea ice drift (SID) observations are used as independent information for the evaluation, additionally to the assimilated OSI-SAF SIC and CS2SMOS SIT data. The subject of the paper is well within the frames of the journal. Generally, the paper is well structured and detailed, and clearly written; the figures are of a good quality; the method used is well justified. However, I have got few comments (*e.g.* on the system settings), which the authors might still want to clarify and further discuss in the manuscript before publishing.

**Specific comments:**

1) Abstract (Lines 13 -14): Please consider rephrasing the last sentence of the Abstract, I am not sure it can be stated in such a generalized context that the "model … demonstrates comparable skills to operational forecasting model**s** that use DA", since there was no explicit comparison to "operational DA forecasting model**s**" carried out in this study and discussed in the paper (except for references to TOPAZ system).

2) I am a bit concerned about the definition of 'bias' in line 320. In line 320 the bias is defined as "model-minus-observations $d(t)=H(x(t)) − y(t))$", while "$d(t)=H(x(t)) − y(t))$" is indeed innovation (a difference between modeled and observed variable). The bias, from the statistical point of view, is an analyzed systematic feature of the innovation after averaging spatially or/and temporally as shown, for instance, in Figure 2b and presented in Table 2.

3) Equation 2: Please double check whether the formulated in the equation is correct and whether it is what has been implemented in the study to approximate the SIT uncertainties. Given equation 2 the observational error variance is a discontinues function of sea ice thickness (SIT, $h_{ice}$): with too (unrealistically) strong increase with $h_{ice}$ for the $h_{ice}$ less than 3 m and saturated small (too small?) values for $h_{ice}$ larger than 3 m (see Figure R1).

[Figure]

*Figure R1: The assumed SIT observational variance as function of SIT reconstructed given Equation 2.*

4) Inflation (Lines 251 - 255). Necessity of inflation was emphasized also in other studies dealing with real sea ice thickness observations. Especially, it was required when no forcing perturbation was used. (More references could be added). Please elaborate a bit more on this ("Inflation") step of the data assimilation: if/how it relates to forcing perturbation; how the regular inflation within DEnKF works; and why it was additionally required to increase by factor of two the observational variance (it means all the assumed data uncertainties (Eq.1 and Eq.2) were further increased).

   a. Are there any other arguments to increase the assumed observational errors? Representation error? Possible misrepresentation of observational errors by Eq. 1 and Eq. 2 (Figure R1 and Figure R2a)?

   b. Whether the finally considered SIC uncertainty (as a result of doubled SIC observational variance, Figure R2B) is not too large to properly constraint the model if the observed SIC is 0.5±0.2; could it be one of the reasons of "moderate extent" of the SIC improvement?

[Figure]

*Figure R2: Assumed SIC observational variance as a function of SIC, reconstructed given equation 1 (a) and final SIC uncertainty as a result of doubled SIC observational variance (b).*

   c. Were there any sensitivity experiments caried out with respect the original ensemble spread due to perturbation of the atmospheric and oceanic forcing and the internal model parameter?

**Minor comments**

Line 161: why 2.5 km/2days not 1.25 km/day, could it be better to convert to and use m/s units

Line 483, 485: similar comment on the "km/2days" used as units for velocity while m/s is used few lines above. I understand that the authors would like to refer somehow to the decorrelation time scale, nevertheless, I still think that m/s would be a more meaningful unit.

**Typo/misprints**

Line 87: citation format issue – missing reference
Line 92: citation format issue – missing reference
Line 143: version 2o3
Line 278: a space required after the dot in "run.Especially"

---

## Referee Comment (RC2)

Review of "Novel Arctic sea ice data assimilation combining ensemble Kalman filter with a Lagrangian sea ice model" by S. Cheng *et al.* submitted to "The Cryosphere"

Review by François Massonnet (francois.massonnet@uclouvain.be)
The authors can contact me if clarifications are required.

Note: I have not read the other referee's report before writing mine, for the sake of independency.

In this study, Cheng and co-authors document the results from a series of numerical experiments conducted with the Lagrangian sea ice model neXtSIM and constrained by the data assimilation (DA) of sea ice concentration (SIC) and/or sea ice thickness (SIT) at two temporal frequencies (1 day or 7 days) with a deterministic Ensemble Kalman Filter. Specifically, the paper reports a case study over winter 2019-2020. The inherent difficulty of performing data assimilation of Eulerian observations in a Lagrangian model is overcome by remapping the models output before the DA step. Different aspects of model performance are discussed.

The novelty of the manuscript resides, I believe, in the first-ever application of an advanced DA method to a Lagrangian sea ice model (note that I am not aware whether DA methods have ever been applied to Lagrangian ocean models, but that could be worth a quick literature review to also position this paper with the literature on that regard). Therefore, I think the paper is eventually suitable for publication. However, I have several concerns and questions that I would like to raise and see addressed, before the manuscript is accepted.

First, I remember from discussion with several authors some years ago, that clean DA with Lagrangian models posed extremely challenging methodological questions as it is clearly not obvious to update an ensemble when each forecast "lives" on its own mesh. I somehow understand that the approach followed here is a fallback solution (and that's perfectly valid), but I think then that it would be useful for the readership to report on the negative results, what has been tried, etc. so that other groups know that this is not an easy problem. Along the same line, I'm a bit skeptical about the title "Novel Arctic sea ice data assimilation…" because many other groups follow exactly the same approach, namely, remapping the model output to observational space before doing the assimilation. I think the novelty here is to use a new type of sea ice model, hence, I would suggest to place the "novel" besides "Lagrangian" (or to drop it).

Second, as it stands, the manuscript does not offer much physical insights, an aspect that I would expect to feature in a journal like The Cryosphere (otherwise the paper is fine for GMD). Currently, the paper is outlined mostly as a development work: there is a Lagrangian model, a data assimilation method, the two are put together, and measures of performance indicate that the DA works. It would be good, not only for the authors but surely for the

entire community, to understand what causes those improvements or lack of improvements. For example, I would have expected to see maps of correlations (in ensemble space) of SIC x SIT, to understand why the SIC7 experiment does not reduce the SIT biases (Fig. 7, row 4), a somewhat surprising result given that other studies have suggested that such cross-improvements are possible. I have seen that this map of correlation was "not shown" in the discussion, so I suspect that the authors have it. Another example: Fig. 6a demonstrates the added value of SIT assimilation. In view of the maps in Fig. 7, it looks like this is possible thanks to a basin-wide reduction of SIT but also to a better representation of sea ice in the transpolar drift area. Do we know why the free model overestimates thickness initially? Can we quantify the process that the DA corrects for, based on those maps? A few more diagnostics (e.g., volume of sea ice created after the assimilation, see e.g. (Mathiot et al., 2012)) would be useful in that respect.

Third, the study is based on experiments spanning half a year (October 2019-2020). I understand that there is an inherent constraint imposed by SIT data unavailability during summer months. Nevertheless, SIC is available throughout the year and this is precisely in May to September months (see, e.g. Fig 1b of (Massonnet et al., 2015)) that SIC assimilation alone might benefit SIT state estimation. The paper would really gain in impact if the SIC7 simulation would be extended until 17 October 2020. I know that it will be difficult (if not impossible) to verify the impact on SIT due to the lack of data during the melting season, but even a SIC performance analysis would be welcome. More generally, it would be good if the paper could cover more than one full annual cycle to make sure that the results are not specific to that 2019-2020 winter. I would propose to amend the title by adding "winter 2019-2020" if the authors choose to not perform those extra experiments.

Fourth, one of the advantages of neXtSIM is its rheology and I am wondering why the paper does not report on any deformation-like metrics, or on linear kinematic features density, etc. Given the improvements in simulated SIT, one could expect such metrics to be better in the DA experiments involving SIT assimilation.

Finally, I have made a few other points (below). I would encourage the authors to implement these changes (along with those mentioned above) to make the manuscript more impactful and relevant for a wide readership. Currently, my own feeling is that it sometimes resembles more a technical report with interesting results, than a paper immediately ready for publication.

Other points
- Line 12: please clarify/rephrase what is meant by "bivariate improvements between SIC and SIT"
- Line 55: "predicting sea ice is more of a boundary condition than an initial value problem". I would be a bit more cautious here. I assume that by "boundary" the authors mean "atmospheric forcing", but in fact, "boundary condition" may mean external forcing from a climate point of view. See, e.g., Blanchard-Wrigglesworth et al. (2011).

- Line 85-87: "The neXtSIM model [...] shows remarkable performance": while I'm convinced that neXtSIM is a great model, the use of "remarkable" is somehow outside what can be expected from a scientific text.
- In relation to the previous comment, I find it odd that the authors cite the Hutter et al. (2022) article but not the companion paper (Bouchat et al., 2022). As I understand the two papers, the Bouchat et al. contribution demonstrates that sea ice model performance for deformation rates is rather independent from the underlying rheological assumptions; while the Hutter al. contribution shows superior performance of the MEB rheology (on which neXtSIM is based) for linear kinematic features. That illustrates that the notion of "performance" always bears some degree of subjectivity – at least in the choice of metric – so that some nuance is always beneficial.
- Line 111-115: The model is forced by an output from IFS, but I could not decide based on the text whether this IFS output is constrained by observations or not. I assume it is since the authors attempt to reproduce observed sea ice conditions for a particular winter. Are the authors then using the output of the ERA5 reanalysis, based on IFS? Clarifications would be welcome.
- Line 172: What exactly does "the sea ice model is nonlinear" mean? Nonlinear in what input?
- Line 180: Regarding the perturbations: I understand that these perturbations bear a spatio-temporal covariance structure, which is a good choice. But do they also bear covariance across variables? Also on that point, why are short-wave radiation, 2m dewpoint temperature, mean sea level pressure, and liquid precipitation not perturbed as well?
- Line 209: add "during the winter season" because the melting can be driven by the atmospheric forcing during spring and summer.
- Line 210: "Recalling…" is not a sentence.
- Line 214: I'm unclear what is the treatment of snow in the model after the DA step, and why it is not included in the list of updated variables. Snow is an important physical parameter that sets the conduction fluxes in winter. If the SIC and / or SIT biases are corrected but the snow depth is left unchanged, it can cause sub-optimal performance of the assimilation, I think. Please clarify this point.
- Line 225: Regarding the mapping procedure, it would be good to know how much interpolation error this procedure introduces to the assimilated fields. One way to do this would be to make a 'dry run', i.e., (1) take the SIC and SIT of one member, (2) interpolate them to the observed grid, (3) interpolate them back to the native member mesh, (4) compute statistics between the original SIC and SIT fields and the SIC and SIT fields that have undergone the back-and forth interpolation. To what extent can this interpolation error be included in the DA uncertainty specifications?
- Line 255: What is the physical basis for a radius of localization of 300 km? Does this correspond to a typical scale of spatial variability for SIC, SIT, or both? Please review the studies of, e.g., Blanchard-Wrigglesworth & Bitz (2014) and Lukovich & Barber (2007)
- Line 279: While I understand the principle, I'm unclear how in practice the consistency check is done. Could you be more specific (or document the code in the Appendix?)

- Line 306: What does "noticing that the spread saturates for ensemble sizes above 40" mean? I'd be surprised that nothing more can be learned by adding more ensemble members. The sentence seems to imply that a 41$^{st}$ ensemble member would necessarily be a linear combination of the first 40, but that's not what I think the authors mean.
- Line 310: "quasi-independent". I'm not so sure about this, because the IFS output forcing the model has been run with prescribed SIC (and possibly SIT? Not sure) conditions, so that when IFS output forces neXtSIM, it re-introduces observed sea ice information although implicitly.
- Line 372: "expected from Lisaeter et al. 2003": please clarify or re-explain why this is expected.
- Fig. 4: A few readers won't be clear how to interpret positive values for underestimation (Fig. 4d). I would clarify this in the caption.
- Line 396: "This is typical of a 'healthy' ensemble that the ensemble forecasts and their ensemble mean are statistically undistinguishable". I would tend to think that in any sequence of number from any distribution, the mean could also be one of the numbers itself (except pathological cases like bi-modal distributions). My (perhaps biased) idea of a healthy ensemble is that the ensemble spread is comparable to the innovations. Clarifications would be good here on what the others really mean.
- Fig. 4: could you please ensure that the y-axis limits are the same, for easy comparison?
- Line 437: is there a way to know why the joint assimilation of SIC & SIT degrades the SIT compared to the SIT assimilation? Related to one of my main comments, some physical understanding going beyond the description of the result would bring value to the paper.
- Line 449: "the relationship between the two variables is nonlinear" → can you clarify? By showing a scatter plot, for example?
- Fig. 8. The figure is, sincerely, very difficult to interpret because the curves are so close to each other. Isn't there a more effective way to demonstrate the impact of the DA on the simulated drift? Did you consider, for example, showing histograms of ice drift bias instead of time series? I'm not sure the temporal aspect is particularly important here since no obvious seasonality emerges.

Typos
- Line 17: add space before parenthesis
- Line 19: forecast → forecasts
- Line 49-50: "the observations […] observe" is redundant
- Line 60: "construct ensemble" → "construct an ensemble" ?
- Line 87: there is a missing reference "?"
- Line 92: same
- Line 278: missing space before "Especially"
- Fig. 4 caption: "Extend" → "Extent"

Blanchard-Wrigglesworth, E., & Bitz, C. M. (2014). Characteristics of Arctic Sea-Ice Thickness

Variability in GCMs. *Journal of Climate*, *27*(21), 8244-8258.

https://doi.org/10.1175/JCLI-D-14-00345.1

Blanchard-Wrigglesworth, E., Bitz, C. M., & Holland, M. M. (2011). Influence of initial

conditions and climate forcing on predicting Arctic sea ice. *Geophysical Research*

*Letters*, *38*(18), n/a-n/a. https://doi.org/10.1029/2011GL048807

Bouchat, A., Hutter, N., Chanut, J., Dupont, F., Dukhovskoy, D., Garric, G., Lee, Y. J., Lemieux,

J.-F., Lique, C., Losch, M., Maslowski, W., Myers, P. G., Ólason, E., Rampal, P.,

Rasmussen, T., Talandier, C., Tremblay, B., & Wang, Q. (2022). Sea Ice Rheology

Experiment (SIREx) : 1. Scaling and Statistical Properties of Sea-Ice Deformation

Fields. *Journal of Geophysical Research: Oceans*, *127*(4), e2021JC017667.

https://doi.org/10.1029/2021JC017667

Hutter, N., Bouchat, A., Dupont, F., Dukhovskoy, D., Koldunov, N., Lee, Y. J., Lemieux, J.-F.,

Lique, C., Losch, M., Maslowski, W., Myers, P. G., Ólason, E., Rampal, P., Rasmussen,

T., Talandier, C., Tremblay, B., & Wang, Q. (2022). Sea Ice Rheology Experiment

(SIREx) : 2. Evaluating Linear Kinematic Features in High-Resolution Sea Ice

Simulations. *Journal of Geophysical Research: Oceans*, *127*(4), e2021JC017666.

https://doi.org/10.1029/2021JC017666

Lukovich, J. V., & Barber, D. G. (2007). On the spatiotemporal behavior of sea ice

concentration anomalies in the Northern Hemisphere. *Journal of Geophysical*

*Research: Atmospheres*, *112*(D13). https://doi.org/10.1029/2006JD007836

Massonnet, F., Fichefet, T., & Goosse, H. (2015). Prospects for improved seasonal Arctic sea

ice predictions from multivariate data assimilation. *Ocean Modelling*, *88*, 16-25.

https://doi.org/10.1016/j.ocemod.2014.12.013

Mathiot, P., König Beatty, C., Fichefet, T., Goosse, H., Massonnet, F., & Vancoppenolle, M.

(2012). Better constraints on the sea-ice state using global sea-ice data assimilation.

*Geoscientific Model Development*, *5*(6), 1501-1515. https://doi.org/10.5194/gmd-5-

1501-2012

---

## Author Comment (AC1)

Author's response:
We thank Reviewer 1 for her/his dedicated comments. We have revised the manuscript accordingly and we hope the current version reaches the high standard expected. We respond point-by-point to the Reviewer's remarks in the following. Our replies are written in blue. Line numbers mentioned in this document are from the revised manuscript with track changes

Review of the manuscript "Novel Arctic sea ice data assimilation combining ensemble Kalman filter with a Lagrangian sea ice model" by Sukun Cheng et al.

The paper presents an experience of implementing a Kalman-type ensemble-based filter to combine sea ice concentration (SIC) and thickness (SIT) observational information with a Lagrangian sea ice model. In particular, the authors assimilate SIC from the Ocean and Sea Ice Satellite Application Facility (OSI-SAF) and the merged SIT product from CryoSat-2 and SMOS satellite missions into the Lagrangian sea ice model neXtSIM with the deterministic Ensemble Kalman filter (DEnKF). The filter analysis is performed on the ensemble of Lagrangian model states individually interpolated to a reference grid. The updated states are projected back onto the temporarily variable model mesh to reinitialize the model for the next forecast phase. The sea ice forecasting system is evaluated for the Arctic Ocean over the 2019/2020 winter time period. The OSI-SAF sea ice drift (SID) observations are used as independent information for the evaluation, additionally, to the assimilated OSI-SAF SIC and CS2SMOS SIT data. The subject of the paper is well within the frames of the journal. Generally, the paper is well structured and detailed, and clearly written; the figures are of a good quality; the method used is well justified. However, I have got few comments (*e.g.* on the system settings), which the authors might still want to clarify and further discuss in the manuscript before publishing.

**Specific comments:**
1) Abstract (Lines 13 -14): Please consider rephrasing the last sentence of the Abstract, I am not sure it can be stated in such a generalized context that the "model ... demonstrates comparable skills to operational forecasting models that use DA", since there was no explicit comparison to "operational DA forecasting models" carried out in this study and discussed in the paper (except for references to TOPAZ system).

We agree with the Reviewer that comparisons with operational DA forecasting models are not quantitatively carried out in the manuscript. This was concluded based on a qualitative comparison with the TOPAZ4 output which we are familiar with.
Following the Reviewer's suggestion, we rephrase the last sentence in the abstract.

2) I am a bit concerned about the definition of 'bias' in line 320. In line 320 the bias is defined as "model-minus-observations $d(t)=H(x(t)) - y(t))$", while "$d(t)=H(x(t)) - y(t))$" is indeed innovation (a difference between modeled and observed variable). The bias, from the statistical point of view, is an analyzed systematic feature of the innovation after averaging spatially or/and temporally as shown, for instance, in Figure 2b and presented in Table 2.

Thank you for pointing out this potential confusion.

Our definition of bias follows from Williams et al. (2021). As we always use bias as spatial or temporal means, we add an average operator in our definition. See line 380 in the revised manuscript.

3) Equation 2: Please double check whether the formulated in the equation is correct and whether it is what has been implemented in the study to approximate the SIT uncertainties. Given equation 2 the observational error variance is a discontinues function of sea ice thickness (SIT, $h_{ice}$): with too (unrealistically) strong increase with $h_{ice}$ for the $h_{ice}$ less than 3 m and saturated small (too small?) values for $h_{ice}$ larger than 3 m (see Figure R1).

We apologise but there was a typo in the equation for the condition of thickness < 3m. We are thankful to the Reviewer for noticing it. The typo has now been corrected by adding a negative sign in the exponent in the second term.

$$\sigma^2_{\text{obs,SIT}} = \begin{cases} \min(0.2, 0.02e^{1.8(h_{ice}-3)}), & h_{ice} > 3m, \\ \max(0.02, 0.1e^{-1.5h_{ice}}), & \text{otherwise.} \end{cases} \qquad (2)$$

4) Inflation (Lines 251 - 255). Necessity of inflation was emphasized also in other studies dealing with real sea ice thickness observations. Especially, it was required when no forcing perturbation was used. (More references could be added). Please elaborate a bit more on this ("Inflation") step of the data assimilation: if/how it relates to forcing perturbation; how the regular inflation within DEnKF works; and why it was additionally required to increase by factor of two the observational variance (it means all the assumed data uncertainties (Eq.1 and Eq.2) were further increased).

Prompted by the Reviewer's suggestion, we address the inflation step more extensively in the revised section 4.5.2 (lines 290-303).

a. Are there any other arguments to increase the assumed observational errors? Representation error? Possible misrepresentation of observational errors by Eq. 1 and Eq. 2 (Figure R1 and Figure R2a)?

Observational errors generally include measurement (instrument) errors and representation errors. The representation errors cover several error sources including unresolved scales and processes, observation-operator errors, and quality-control errors (Janjic et al., 2017).

However, the sea ice products we used only provide parts of the observation errors, which are not sufficient to represent the entire observational errors. In particular, CS2SMOS thickness data provides errors in the merging and interpolation of CryoSat-2 and SMOS products. While the OSISAF ice concentration product provides a concentration algorithm and tie-point uncertainties and smearing uncertainty due to satellite footprint mismatches. This situation motivates the increase of observational errors to properly represent the associated uncertainty.

We quoted a discussion from William et al. (2021) below as a reference: "The error levels in the CS2-SMOS product are only the interpolation error and are thus a lower bound as they do not include uncertainties in the individual CS2 and SMOS products. CS2 in particular is sensitive to the ice and snow densities used or the snow thickness which affect the conversion from freeboard to thickness (Zygmuntowska et al., 2014)."

b. Whether the finally considered SIC uncertainty (as a result of doubled SIC observational variance, Figure R2B) is not too large to properly constraint the model if the observed SIC is 0.5±0.2; could it be one of the reasons of "moderate extent" of the SIC improvement?

[Figure]

*Figure R2: Assumed SIC observational variance as a function of SIC, reconstructed given equation 1 (a) and final SIC uncertainty as a result of doubled SIC observational variance (b).*

This is possible, but not very likely in our case. The spatially averaged sea ice concentration is above 0.75 around mid-November 2019 in Fig. 2(a), which has a smaller observation error compared to the start of November as shown in Fig. R2. With higher sea ice concentration and lower observation uncertainties, the improvements get less significant. This suggests that, instead of misspecified observation uncertainties, the moderate improvements are related to the high sea ice concentration where the majority of the Arctic ocean is covered by ice without much need for improvements by infrequent assimilations where the uncertainties in the forcing propagate to the model forecast.

c. Were there any sensitivity experiments carried out with respect to the original ensemble spread due to perturbation of the atmospheric and oceanic forcing and the internal model parameter?
The sensitivity experiments on the impact of perturbation of the atmospheric forcing and the internal model parameter (ice cohesion) are reported by Cheng et al. (2020). Those experiments have indeed guided and motivated the experimental setup in this current work.

**Minor comments**
Line 161: why 2.5 km/2days not 1.25 km/day, could it be better to convert to and use m/s units
Line 483, 485: similar comment on the "km/2days" used as units for velocity while m/s is used few lines above. I understand that the authors would like to refer somehow to the decorrelation time scale, nevertheless, I still think that m/s would be a more meaningful unit.
In this study, we evaluate the model data on the observation spaces rather than the other way around. Presenting the ice drift data km/2days is to be consistent with the OSISAF ice drift observations settings for comparison. According to the OSI-SAF ice drift introduction https://osisaf-hl.met.no/osi-405-c-desc, the time span of the OSISAF ice drift observations is

48 hours or 2 days. It is the time delay between the start time and the stop time of the ice motion described by one vector.

More importantly, the ice drift velocity varies over time, thus its displacement is nonlinear over 2 days. Averaging the quantities over unit time could cover this fact and cause potentially missing interpolation in the evaluation.

**Typo/misprints**

Line 87: citation format issue – missing reference                                    fixed
Line 92: citation format issue – missing reference                                    fixed
Line 143: version 2o3                              It seems fine to use CS2SMOS version
203 instead of 2o3, according to
https://spaces.awi.de/display/CS2SMOS/CryoSat-SMOS+Merged+Sea+Ice+Thickness
Line 278: a space required after the dot in "run. Especially"                     fixed

**Reference**

Anderson, J. L., & Anderson, S. L. (1999). A Monte Carlo implementation of the nonlinear filtering problem to produce ensemble assimilations and forecasts. Monthly weather review, 127(12), 2741-2758.

Anderson, J. L. (2007). Exploring the need for localization in ensemble data assimilation using a hierarchical ensemble filter. Physica D: Nonlinear Phenomena, 230(1-2), 99-111.

Anderson, J., Hoar, T., Raeder, K., Liu, H., Collins, N., Torn, R., & Avellano, A. (2009). The data assimilation research testbed: A community facility. Bulletin of the American Meteorological Society, 90(9), 1283-1296.

Sakov, P., Counillon, F., Bertino, L., Lisæter, K. A., Oke, P. R., & Korablev, A. (2012). TOPAZ4: an ocean-sea ice data assimilation system for the North Atlantic and Arctic. Ocean Science, 8(4), 633-656.

Zygmuntowska, M., Rampal, P., Ivanova, N., & Smedsrud, L. H. (2014). Uncertainties in Arctic sea ice thickness and volume: new estimates and implications for trends. The Cryosphere, 8(2), 705-720.

Xie, J., Bertino, L., Counillon, F., Lisæter, K. A., & Sakov, P. (2017). Quality assessment of the TOPAZ4 reanalysis in the Arctic over the period 1991–2013. Ocean Science, 13(1), 123-144.

Janjić, T., Bormann, N., Bocquet, M., Carton, J. A., Cohn, S. E., Dance, S. L., ... & Weston, P. (2018). On the representation error in data assimilation. Quarterly Journal of the Royal Meteorological Society, 144(713), 1257-1278.

Cheng, S., Aydoğdu, A., Rampal, P., Carrassi, A., & Bertino, L. (2020, December). Probabilistic forecasts of sea ice trajectories in the Arctic: impact of uncertainties in surface wind and ice cohesion. In Oceans (Vol. 1, No. 4, pp. 326-342). MDPI.

Williams, T., Korosov, A., Rampal, P., & Ólason, E. (2021). Presentation and evaluation of the Arctic sea ice forecasting system neXtSIM-F. The Cryosphere, 15(7), 3207-3227.

---

## Author Comment (AC2)

Author's response:

We thank the Reviewer - François Massonnet - for his careful reading and comprehensive review. We believe that the work has greatly improved thanks to answering the Reviewer's comments. Our point-by-point replies are provided below together with a description of the proposed changes. The Reviewer's comments are listed in black below while our responses are in blue. Line numbers mentioned in this document are from the revised manuscript with track changes.

In this study, Cheng and co-authors document the results from a series of numerical experiments conducted with the Lagrangian sea ice model neXtSIM and constrained by the data assimilation (DA) of sea ice concentration (SIC) and/or sea ice thickness (SIT) at two temporal frequencies (1 day or 7 days) with a deterministic Ensemble Kalman Filter. Specifically, the paper reports a case study over winter 2019-2020. The inherent difficulty of performing data assimilation of Eulerian observations in a Lagrangian model is overcome by remapping the model's output before the DA step. Different aspects of model performance are discussed.

The novelty of the manuscript resides, I believe, in the first-ever application of an advanced DA method to a Lagrangian sea ice model (note that I am not aware whether DA methods have ever been applied to Lagrangian ocean models, but that could be worth a quick literature review to also position this paper with the literature on that regard). Therefore, I think the paper is eventually suitable for publication. However, I have several concerns and questions that I would like to raise and see addressed, before the manuscript is accepted.

First, I remember from discussion with several authors some years ago, that clean DA with Lagrangian models posed extremely challenging methodological questions as it is clearly not obvious to update an ensemble when each forecast "lives" on its own mesh. I somehow understand that the approach followed here is a fallback solution (and that's perfectly valid), but I think then that it would be useful for the readership to report on the negative results, what has been tried, etc. so that other groups know that this is not an easy problem.

We address the comment below. Parts of the following response have been added in the Introduction section of the revised manuscript.

There is an intrinsic challenge in applying ensemble data assimilation (DA) techniques when an adaptive moving mesh (AMM) solver is being used for the model since the member grids vary independently of each other. The natural, straightforward solution is to adopt a reference mesh, on which the mean and error covariances of the ensemble can be calculated. Such a strategy was first, to the best of our knowledge, carried out by Du et al. (2016) for an ocean model on an AMM. Super-meshing techniques were developed by Farrell et al. (2009) and used in the paper by Du et al. (2016) and, in a slightly different way, by Jain et al. (2018). The super-meshing can reduce the interpolation error. At this point, the use of a reference mesh can be considered the standard and time-tested method for dealing with an ensemble each of whose members live on a different grid.

A group that overlaps with that of the current paper authored a new approach that did not involve using a reference mesh. The discussions to which the reviewer anecdotally refers

were at the time this approach was being developed. In this strategy, both the model variables and the grid locations are updated at the assimilation step. Bonan et al. (2017) took this approach in a one-dimensional model of a grounded shallow water ice sheet model. But their model did not involve remeshing and so the strategy of augmenting the state variable with node locations worked in a natural way. In the context of a model with remeshing, Sampson et al. (2021) implemented a method that afforded mesh location updates. These approaches have, however, only been developed for one-dimensional models and it remains a significant challenge to extend them to two or more space dimensions. Particularly for the Lagrangian neXtSIM model, it remains unclear as to how this should even be set up, for instance, whether the cell centers or the vertices should be appended as state variables.

The data assimilation methodology used in this study is the 2D version of the one that is shown to work on 1D AMM models studied by Aydogdu et al. (2019). In that work, some challenges in applying DA to adaptive meshes are briefly reviewed and the different techniques to overcome the issue in the simplified 1D models (Burgers and Kuramoto-Shivasinsky) are discussed. These models are discretized on a 1D adaptive mesh to imitate neXtSIM with lower computational cost. Finally, their methodology proposes an upper and lower limit for the resolution of the fixed reference mesh using the remeshing criteria intrinsic to the AMM methodology. They concluded that a lower-resolution analysis mesh, which has a resolution close to the Lagrangian mesh, improves the analysis almost as well as a high-resolution mesh and if the concern involves also the computational cost it can be preferable to use a lower-resolution fixed reference mesh. This is the basis of the choice for the fixed reference mesh used in this study.

Along the same line, I'm a bit skeptical about the title "Novel Arctic sea ice data assimilation…" because many other groups follow exactly the same approach, namely, remapping the model output to observational space before doing the assimilation. I think the novelty here is to use a new type of sea ice model, hence, I would suggest to place the "novel" besides "Lagrangian" (or to drop it).
We agree to modify the title as "Arctic sea ice data assimilation combining ensemble Kalman filter with a novel Lagrangian sea ice model for the winter 2019-2020".

Nevertheless, there is more nuance here in the remapping procedure. Our approach maps different ensemble members to a fixed regular reference grid. This remapping process is not part of the observation operator. Instead, the observation operator transforms from the fixed reference grid to the observation spaces, which can have both OSISAF and CS2SMOS data. Hence, the DA is performed to generate analysis on a fixed-regular-reference grid, which is then mapped to the unstructured model grid again.

Second, as it stands, the manuscript does not offer much physical insights, an aspect that I would expect to feature in a journal like The Cryosphere (otherwise the paper is fine for GMD). Currently, the paper is outlined mostly as a development work: there is a Lagrangian model, a data assimilation method, the two are put together, and measures of performance

indicate that the DA works. It would be good, not only for the authors but surely for the entire community, to understand what causes those improvements or lack of improvements. We thank the Reviewer for his criticism on this aspect. We have filled the lack of sufficient physical insight into our results, by answering the specific comments/suggestions below, as well as in other instances in the manuscript.

For example, I would have expected to see maps of correlations (in ensemble space) of SIC x SIT, to understand why the SIC7 experiment does not reduce the SIT biases (Fig. 7, row 4), a somewhat surprising result given that other studies have suggested that such cross-improvements are possible. I have seen that this map of correlation was "not shown" in the discussion, so I suspect that the authors have it.

We note that the weak improvement on SIT in the SIC7 experiment is largely due to a constraint in post-processing when updating the state vector in neXtSIM. As explained in Sect. 4.5.3, the constraint is applied when assimilating SIC only and it limits the state update only where the sea ice density is less than 90%. The constraint's impact is illustrated in Figure R1. Specifically about the Reviewer's concern, the left panel of Figure R1 shows maps of the correlation coefficients between SIC and SIT using the forecast ensemble from the SIC7 experiment. The date is 19, Dec 2019, and it is a typical winter day in the Arctic Oceans. The right panel shows the same coefficients after applying a mask to show only the locations where ice concentration is less than 90%. This reveals a much smaller area that is partially ice-covered. Recall that, due to the aforementioned constraint, DA only applies to model states on the grid cells in the right panel of Figure R1, making the DA impact in the SIC7 experiment very limited.

The overall improvement on SIT in the SIC7 experiment over the Free Run was not significant compared with other DA experiments where SIT is assimilated. Nevertheless, Fig. 6(b) of the manuscript indicates that the spatial averaged SIT bias in the SIC7 is lower than that of the Free Run. The relative reduction reaches up to 25% of the Free Run's bias at the end of the simulated period. In addition, we highlight the improvement in local areas. For example, in Kara and Barents Seas, the overestimated thickness in the Free Run is mitigated or almost absent in SIC7.

[Figure]

Figure R1 (left) Maps of correlation coefficients between SIC and SIT using forecast ensemble from the SIC7 experiment on 19, Dec. 2019. (right) same as the left panel but applied a mask of concentration<=90%.

Another example: Fig. 6a demonstrates the added value of SIT assimilation. In view of the maps in Fig. 7, it looks like this is possible thanks to a basin-wide reduction of SIT but also to a better representation of sea ice in the transpolar drift area. Do we know why the free model overestimates thickness initially?

We thank the Reviewer for this insightful comment. The version of the model we used in this study had an incorrect ridging parameter, which prevented the sea ice export in the Fram Strait and accumulated ice thickness in the important areas in the North of Greenland and North of Svalbard. This parameter has been corrected posterior to our study and the change has been documented in Boutin et al. (2022).

Can we quantify the process that the DA corrects for, based on those maps? A few more diagnostics (e.g., volume of sea ice created after the assimilation, see e.g. (Mathiot et al., 2012))

Figure R2(a) shows the daily evolution of sea ice volume (SIV) during winter 2019-2020 among the experiments with assimilation. It also superimposes the SIV observations from CS2SMOS data. The relationship of SIV between the model predictions and observations is similar to that of SIT in Figure 6(a) and refer to the relevant SIT discussion in the manuscript. The increments of SIV by the DEnKF are shown in panel (b), which indicates the DA corrects on SIV. The signs of increment valid that DA can effectively bring the modeled SIV to the observations. In the SIC1-SIT7 experiment, the increments by assimilating SIC each day are generally negative, but large positive increments are observed every 7 days when SIT is assimilated. It indicates the correction of SIV is more related to DA of SIT when SIC in the Arctic is mostly close to 1.

[Figure]

Figure R2. Time series of (a) daily total SIV over the Arctic domain and (b) increment of SIV by DEnKF during winter 2019-2020.

Third, the study is based on experiments spanning half a year (October 2019-2020). I understand that there is an inherent constraint imposed by SIT data unavailability during summer months. Nevertheless, SIC is available throughout the year and this is precisely in May to September months (see, e.g. Fig 1b of (Massonnet et al., 2015)) that SIC assimilation alone might benefit SIT state estimation. The paper would really gain in impact if the SIC7

simulation would be extended until 17 October 2020. I know that it will be difficult (if not impossible) to verify the impact on SIT due to the lack of data during the melting season, but even a SIC performance analysis would be welcome.

More generally, it would be good if the paper could cover more than one full annual cycle to make sure that the results are not specific to that 2019-2020 winter.

We focus on investigating the best EnKF strategies for the neXtSIM model, including the multivariate assimilation between SIT and SIC. Thus, we choose the winter months in which the observations of both variables are available. Selecting the 2019-2020 winter is motivated by a prior study of neXtSIM with a simple DA method (Williams et al., 2021). They reported a significant overestimation of thickness in the 2019-2020 winter.

Another reason for focusing on the winter is the presence of Linear Kinematic Features (LKFs). We would like to assimilate the position of LKFs in a follow-up study that will re-use the same framework. LKFs are mostly absent in the summer and we have less interest in processing that time. Running a summer situation would complicate the narrative of the paper and we prefer to focus on winter.

We agree that a longer period of runs would have made a more convincing demonstration to the readers. With a recently released year-round Cryosat-2 thickness product (https://www.nature.com/articles/s41586-022-05058-5), we plan for new experiments for multi-year simulations in future work.

I would propose to amend the title by adding "winter 2019-2020" if the authors choose to not perform those extra experiments.

We have adopted the suggestion to add "winter 2019-2020" at the end of the title as "Arctic sea ice data assimilation combining ensemble Kalman filter with a novel Lagrangian sea ice model for the winter 2019-2020"

Fourth, one of the advantages of neXtSIM is its rheology and I am wondering why the paper does not report on any deformation-like metrics, or on linear kinematic features density, etc. Given the improvements in simulated SIT, one could expect such metrics to be better in the DA experiments involving SIT assimilation.

A complete understanding of the impact of ensemble assimilation on sea-ice forecasts from neXtSIM is a large project. Considering the time and resources available, we decided to focus in this paper on the improvement of the main sea ice features by assimilation and we will continue to investigate more sea ice features including ice deformation due to DA over longer time periods in future work.

However, what we can say from our experience evaluating sea ice dynamics in models, is that the changes on SIT made by our assimilation procedure are not large enough to significantly influence the scaling properties of the simulated sea ice deformation, meaning that the differences are not expected to be statistically significant.

Finally, I have made a few other points (below). I would encourage the authors to implement these changes (along with those mentioned above) to make the manuscript more impactful and relevant for a wide readership. Currently, my own feeling is that it sometimes resembles more a technical report with interesting results, than a paper immediately ready for publication.

Thank you.

Other points
• Line 12: please clarify/rephrase what is meant by "bivariate improvements between SIC and SIT"

We meant SIT-SIC mutual improvements when assimilating one gives information on the other and vice-versa. Also in light of the previous points on the SIT-SIC relation, we decided to remove the sentence.

• Line 55: "predicting sea ice is more of a boundary condition than an initial value problem". I would be a bit more cautious here. I assume that by "boundary" the authors mean "atmospheric forcing", but in fact, "boundary condition" may mean external forcing from a climate point of view. See, e.g., Blanchard-Wrigglesworth et al. (2011).

We agree with the reviewer. This cited part is rephrased: see line 65 of the revised manuscript.

• Line 85-87: "The neXtSIM model [...] shows remarkable performance": while I'm convinced that neXtSIM is a great model, the use of "remarkable" is somehow outside what can be expected from a scientific text.

We replace "remarkable" with "good", which is on lines 119-120 of the revised manuscript.

• In relation to the previous comment, I find it odd that the authors cite the Hutter et al. (2022) article but not the companion paper (Bouchat et al., 2022). As I understand the two papers, the Bouchat et al. contribution demonstrates that sea ice model performance for deformation rates is rather independent from the underlying rheological assumptions; while the Hutter al. contribution shows superior performance of the MEB rheology (on which neXtSIM is based) for linear kinematic features. That illustrates that the notion of "performance" always bears some degree of subjectivity – at least in the choice of metric – so that some nuance is always beneficial.

We thank the reviewer for mentioning these important works. Although we did not cite these two papers in the original manuscript, we are glad to mention them in section 2.1 of the latest manuscript (lines 120-124).

• Line 111-115: The model is forced by an output from IFS, but I could not decide based on the text whether this IFS output is constrained by observations or not. I assume it is since the authors attempt to reproduce observed sea ice conditions for a particular winter. Are the authors then using the output of the ERA5 reanalysis, based on IFS? Clarifications would be welcome.

The atmosphere forcing we used is ECMWF IFS (cycle 45r1) analysis product with lead time 0, not the ERA5 reanalysis. Thanks for pointing out this potential confusion. We have clarified this point in the manuscript (line 149).

• Line 172: What exactly does "the sea ice model is nonlinear" mean? Nonlinear in what input?

We meant the time-evolving constitutive equations of the model. However, to avoid confusion, we have removed the sentence.

• Line 180: Regarding the perturbations: I understand that these perturbations bear a spatio-temporal covariance structure, which is a good choice. But do they also bear covariance across variables? Also on that point, why are short-wave radiation, 2m dewpoint temperature, mean sea level pressure, and liquid precipitation not perturbed as well?

The perturbations of wind velocity are derived from the sea level pressure with a non-divergent constraint, while the other perturbed variables are perturbed independently. There is no constraint from the covariance across those variables.

The precipitation during winter could be approximated by the snowfall. From the perspective of data assimilation, for state estimate, it is ideal to limit the source of uncertainty one wishes to mimic within the ensemble, to those that actually contribute to the forecast error. This is of course not possible in practice, for many reasons, including lack of sufficient spread. Thus our pragmatic choice has been to inquire neXtSIM model developers on which inputs have, in their opinion, the larger uncertainty. On that basis we have decided what variables to perturb. The unperturbed terms are the dew point and shortwave radiation, which could be perturbed in a similar way in the future, although current efforts are going in the direction of using the ECMWF ensemble forecasts instead of (or in combination with) random perturbations.

• Line 209: add "during the winter season" because the melting can be driven by the atmospheric forcing during spring and summer.

Thank you for your comments. The sentence is modified in the latest manuscript (line 236) as follows.

"Moreover, because the freezing and melting of sea ice are strongly driven by the ocean boundary conditions during the winter season, we also include the SSS and SST among the analyzed variables."

• Line 210: "Recalling..." is not a sentence.

We apologise for the grammar error that is now fixed. Thank you.

• Line 214: I'm unclear what is the treatment of snow in the model after the DA step, and why it is not included in the list of updated variables. Snow is an important physical parameter that sets the conduction fluxes in winter. If the SIC and / or SIT biases are corrected but the snow depth is left unchanged, it can cause sub-optimal performance of the assimilation, I think. Please clarify this point.

It is indeed possible to include the snow depth variable in the state vector, as has always been practiced in the TOPAZ system, however, we have never tested how important its inclusion is. In this study, the neXtSIM model is much more novel than in TOPAZ and we have a priori decided not to include the snow depth as a matter of safety.

Although the snow thickness is not included in the analysis, we update this variable in the post-process according to the change of concentration.

- Line 225: Regarding the mapping procedure, it would be good to know how much interpolation error this procedure introduces to the assimilated fields. One way to do this would be to make a 'dry run', i.e., (1) take the SIC and SIT of one member, (2) interpolate them to the observed grid, (3) interpolate them back to the native member mesh, (4) compute statistics between the original SIC and SIT fields and the SIC and SIT fields that have undergone the back-and-forth interpolation. To what extent can this interpolation error be included in the DA uncertainty specifications?

As the Reviewer suggests, we performed a "dry run" by using one member from the SIT7 experiment. We pick up 26 Dec. 2022 for the calculation when the Arctic basin is mainly covered by sea ice from the left column of Figure R3. The top row indicates SIC and the bottom row indicates SIT, respectively. SIC and SIT of a member are interpolated on the fixed-regular-reference grid. The map of error (the original field minus the back-and-forth interpolated field) is shown in the middle column. The corresponding density distribution of the absolute errors are given in the right column, which show that errors of most grid cells are below 0.01 in SIC and a little bit more errors in SIT that is less than 0.1. The concentration errors originate primarily from the ice edge, while the thickness errors mostly from the multi-year ice areas. Errors are generally small, since the resolutions of the neXtSIM irregular grid and the fixed regular reference grid have similar resolutions in this study. The interpolation errors can be further reduced by increasing the reference grid's resolution (see Aydogdu et al., 2019).

[Figure]

Figure R3. back-and forth interpolation errors calculated using data from SIT experiment on 26, Dec. 2019. (left) field distribution (middle) errors of original field minus the field undergone the back-and-forth interpolation (right) density distribution of the errors for quantities: (a) ice concentration (b) ice thickness.

• Line 255: What is the physical basis for a radius of localization of 300 km? Does this correspond to a typical scale of spatial variability for SIC, SIT, or both? Please review the studies of, e.g., Blanchard-Wrigglesworth & Bitz (2014) and Lukovich & Barber (2007)

We choose the DA settings following the TOPAZ4 EnKF settings, which uses 300 km after some empirical tuning. A small radius has the advantages of increasing the ensemble spread and making the assimilation more efficient, but it should include more observations than the effective number of degrees of freedom of the local system (see the discussion in Sakov and Bertino (2011)). This is achieved with a 300 km radius (90 km effective radius) and an observation grid of 25 km resolution.

• Line 279: While I understand the principle, I'm unclear how in practice the consistency check is done. Could you be more specific (or document the code in the Appendix?)

Because the procedure includes lots of technical treatments, we summarize the main contents of the consistency check below. For more details please refer to the neXtSIM code via this link on GitHub.

https://github.com/nansencenter/nextsim/blob/IOPerturbation/model/finiteelement.cpp#L14520.

The last paragraph of section 4.5.3 in the latest manuscript (lines 290-303) is extended based on this response.

1. The total ice concentration from the DA analysis update should be at [0,1]. If the total ice concentration exceeds 1, the additional fraction will be removed from the young ice category. The corresponding ice thickness and snow thickness will be scaled with the young ice concentration.

2. If the concentration of old ice is less than the minimum threshold, the old ice category is removed: the ice concentration, ice thickness, and snow thickness of the old ice category are set to zero and the temperature is set to the freezing point of the new ice.

3. If the total ice concentration is less than the minimum threshold, the ocean surface is considered as open water. It sets the ice thickness and snow thickness of young ice types, as well as the volume ratio of ridge ice to 0 and the temperature to the freezing point of water.

4. If new ice appears after assimilation, the temperature at the snow-ice interface is first calculated using the zero-layer model, and then the temperatures at the base and top of the ice are obtained by interpolation.

5. Finally, the volume ratio of the ridged ice was constrained to [0,1]. The sea ice concentration, snow, and ice thickness of young and old ice were constrained to positive values.

• Line 306: What does "noticing that the spread saturates for ensemble sizes above 40" mean? I'd be surprised that nothing more can be learned by adding more ensemble members. The sentence seems to imply that a 41st ensemble member would necessarily be a linear combination of the first 40, but that's not what I think the authors mean.

The Reviewer is right. The original statement was misleading. We had indeed tried ensemble sizes up to 100 in previous experiments and found only very little amelioration when for ensemble sizes larger than 40. This behavior is not surprising and it is typical of deterministic EnKFs like the one we used here (see Carrassi et al., 2022 for the rationale).
The sentence is modified in lines 360-363 of the revised manuscript with track changes.

• Line 310: "quasi-independent". I'm not so sure about this, because the IFS output forcing the model has been run with prescribed SIC (and possibly SIT? Not sure) conditions, so that when IFS output forces neXtSIM, it re-introduces observed sea ice information although implicitly.
The ice observations have been used in generating IFS outputs, as well as the TOPAZ outputs. These outputs are used to drive neXtSIM simulation so that ice observations are re-introduced inevitably.

We see the Reviewer has been confused by the phrase 'quasi-independent'. We want to highlight the comparison between the observed and the modeled sea ice states before the analysis step in DA. The relevant part has been rephrased in lines 366-369 of the revised manuscript.

• Line 372: "expected from Lisaeter et al. 2003": please clarify or re-explain why this is expected.
This was an inappropriate citation. We apologise for this and have removed it from the revised manuscript. In the revised manuscript (line 434), we explain that the increase of SIC ensemble spread in Fig.2(d) is caused by greater variability of the modeled SIC among ensemble members during the seasonal ice melt starting from mid-March 2020.

• Fig. 4: A few readers won't be clear how to interpret positive values for underestimation (Fig. 4d). I would clarify this in the caption.
We rephrase the caption of Fig. 4 to clarify this point in the revised manuscript. According to the definition of the quantity 'underestimation of IIEE' by Eq.(3), it is an absolute error indicating the amount of underestimation of ice extent by the model. So underestimation is always positive by definition, even if this is not intuitive.

• Line 396: "This is typical of a 'healthy' ensemble that the ensemble forecasts and their ensemble mean are statistically undistinguishable". I would tend to think that in any sequence of number from any distribution, the mean could also be one of the numbers itself (except pathological cases like bi-modal distributions). My (perhaps biased) idea of a healthy ensemble is that the ensemble spread is comparable to the innovations. Clarifications would be good here on what the others really mean.
We are afraid the statement in the manuscript was confusing and potentially wrong. Thanks for pointing it out. A sign of health in the ensemble is when the truth is indistinguishable from the ensemble members, which is almost the same to say (how the Reviewer says) that the spread is comparable to the innovation. We substitute "ensemble mean" with "the observations" in the latest manuscript (line 458) since we do not have the "truth" here.

• Fig. 4: could you please ensure that the y-axis limits are the same, for easy comparison?
The y-axis in Fig. 4 has been adjusted to be the same.

• Line 437: is there a way to know why the joint assimilation of SIC & SIT degrades the SIT compared to the SIT assimilation? Related to one of my main comments, some physical understanding going beyond the description of the result would bring value to the paper.
We have a discussion about the degraded performance by joint assimilation of SIC and SIT between lines 583-593 in the revised manuscript. In addition, we view the joint assimilation of SIC and SIT as a multi-variable optimization problem. The optimal solution is the best compromise and fits each term less closely individually.

• Line 449: "the relationship between the two variables is nonlinear" à can you clarify? By showing a scatter plot, for example?
The nonlinearity can be seen from a scatter plot between concentration and thickness below, which superimposes the SIC-SIT of each pixel over the Arctic basin on 19, Dec. 2019, from the experiment SIC7.

[Figure]

Figure R6. Scatter plot of SIC and SIT over the Arctic basin on 19, Dec. 2019.

• Fig. 8. The figure is, sincerely, very difficult to interpret because the curves are so close to each other. Isn't there a more effective way to demonstrate the impact of the DA on the simulated drift? Did you consider, for example, showing histograms of ice drift bias instead of time series? I'm not sure the temporal aspect is particularly important here since no obvious seasonality emerges.
We agree that the time series is not more informative than the numbers in the table. Therefore, Fig. 8 is removed.

We calculate the probability densities of bias, RMSD, and VRMSD using the Kernel density estimation function in the Python Pandas library. The corresponding density curves of all experiments are given below. From left to right, the panels are  the probability densities of bias, RMSD, and VRMSD, respectively for the experiments with DA every 7 days. We observe

that the *SIC7* experiment and the free run have similar with larger errors; *SIT7* and *SIC7-SIT7* experiments are closer as a cluster with lower errors than the free run and *SIC7*.
We interpret the results for RMSD and VRMSD that are in agreement with the spatiotemporal average given in Table 2.

[Figure]

Figure R7. The probability densities of (left)bias, (middle) RMSD, and (right)VRMSD, respectively, estimated by the time series in Fig. 8 in the original manuscript.

Typos
• Line 17: add space before parenthesis                                   done
• Line 19: forecast à forecasts                                           done
• Line 49-50: "the observations [...] observe" is redundant              change to "The multivariate aspect will not appear in the present study because we assimilate the two most important model variables (SIC and SIT) from the observations that both have complete spatial coverage."
• Line 60: "construct ensemble" à "construct an ensemble" ?              change to "construct ensembles"
• Line 87: there is a missing reference "?"                               fixed
• Line 92: same                                                          fixed
• Line 278: missing space before "Especially"                           fixed
• Fig. 4 caption: "Extend" à "Extent"                                    fixed

**Reference**

Lukovich, J. V., & Barber, D. G. (2007). On the spatiotemporal behavior of sea ice concentration anomalies in the Northern Hemisphere. Journal of Geophysical Research: Atmospheres, 112(D13).

Farrell, P., Piggott, M., Pain, C., Gorman, G., and Wilson, C.: Conservative interpolation between unstructured meshes via supermesh construction, Comput. Meth. Appl. Mech. Eng., 198, 2632–2642, https://doi.org/10.1016/j.cma.2009.03.004, 2009.  a

Sakov, P., & Bertino, L. (2011). Relation between two common localisation methods for the EnKF. Computational Geosciences, 15(2), 225-237.

Stroeve, J., Hamilton, L. C., Bitz, C. M., & Blanchard-Wrigglesworth, E. (2014). Predicting September sea ice: Ensemble skill of the SEARCH sea ice outlook 2008–2013. Geophysical Research Letters, 41(7), 2411-2418.

Du, J., Zhu, J., Fang, F., Pain, C. C., & Navon, I. M. (2016). Ensemble data assimilation applied to an adaptive mesh ocean model. International Journal for Numerical Methods in Fluids, 82(12), 997-1009.

Bonan, B., Nichols, N. K., Baines, M. J., & Partridge, D. (2017). Data assimilation for moving mesh methods with an application to ice sheet modelling. Nonlinear Processes in Geophysics, 24(3), 515-534.

Jain, P. K., Mandli, K., Hoteit, I., Knio, O., & Dawson, C. (2018). Dynamically adaptive data-driven simulation of extreme hydrological flows. Ocean Modelling, 122, 85-103.

Sampson, C., Carrassi, A., Aydoğdu, A., & Jones, C. K. (2021). Ensemble Kalman filter for nonconservative moving mesh solvers with a joint physics and mesh location update. Quarterly Journal of the Royal Meteorological Society, 147(736), 1539-1561.

Williams, T., Korosov, A., Rampal, P., & Ólason, E. (2021). Presentation and evaluation of the Arctic sea ice forecasting system neXtSIM-F. The Cryosphere, 15(7), 3207-3227.

Boutin, G., Ólason, E. Ö., Rampal, P., Regan, H., Lique, C., Talandier, C., ... & Ricker, R. (2022). Arctic sea ice mass balance in a new coupled ice-ocean model using a brittle rheology framework. The Cryosphere Discussions, 1-31.

Bouchat, A., Hutter, N., Chanut, J., Dupont, F., Dukhovskoy, D., Garric, G., ... & Wang, Q. (2022). Sea Ice Rheology Experiment (SIREx): 1. Scaling and Statistical Properties of Sea-Ice Deformation Fields. Journal of Geophysical Research: Oceans, 127(4), e2021JC017667.

Hutter, N., Bouchat, A., Dupont, F., Dukhovskoy, D., Koldunov, N., Lee, Y. J., ... & Wang, Q. (2022). Sea Ice Rheology Experiment (SIREx): 2. Evaluating Linear Kinematic Features in High-Resolution Sea Ice Simulations. Journal of Geophysical Research: Oceans, 127(4), e2021JC017666.